# Epanechnikov Variational Autoencoder

## Abstract

In this paper, we bridge Variational Autoencoders (VAEs) (Kingma & Welling, 2013) and kernel density estimations (KDEs) (Rosenblatt, 1956; Parzen, 1962) by approximating the posterior by KDEs and deriving a new lower bound of empirical log likelihood. The flexibility of KDEs not only addresses the limitations of Gaussian latent space in vanilla VAE but also provides a new perspective of estimating the KL-divergence term in original evidence lower bound (ELBO). We then propose the Epanechnikov kernel based VAE and show that the Epanechnikov kernel gives the tightest upper bound in estimating the KL-divergence under appropriate conditions (Epanechnikov, 1969; Bickel & Rosenblatt, 1973) in theory. Compared with Gaussian kernel, Epanechnikov kernel has compact support which should make the generated sample less blurry. The implementation of Epanechnikov kernel in VAE is straightforward as it lies in the "location-scale" family of distributions where reparametrization tricks can be applied directly. A series of experiments on benchmark datasets such as MNIST, Fashion-MNIST, CIFAR-10 and CelebA illustrate the superiority of Epanechnikov Variational Autoenocoder (EVAE) over vanilla VAE and other baseline models in the quality of reconstructed images, as measured by the FID score and Sharpness (Tolstikhin et al., 2018).

## 1 Introduction

In variational inference, an autoencoder learns to encode the original data $\mathbf{x}$ using learnable function $f_\phi^{-1}(\mathbf{x})$ and then to reconstruct $x$ using a decoder $g_\theta$. Kingma & Welling (2013) proposed a stochastic variational inference and learning algorithm called Variational Autoencoders (VAEs) which can be further used to generate new data. According to VAE, a lower bound on the empirical likelihood, which is known as ELBO, is maximized so that the fitted model $p_\theta(\mathbf{x}|\mathbf{z}) = \mathcal{N}(\mathbf{x}; g_\theta(\mathbf{x}), I_D)$ and $q_\phi(\mathbf{z}|\mathbf{x}) = \mathcal{N}(\mathbf{x}; f_\phi(\mathbf{x}), I_D)$ can approximate the true conditional distributions $p(\mathbf{x}|\mathbf{z})$ and $p(\mathbf{z}|\mathbf{x})$, respectively. The isotropic Gaussian prior and posterior distribution in vanilla VAE is mathematically convenient since the corresponding ELBO is analytic. But the main drawbacks are the lack of expressibility of latent space and the possible posterior collapse. There are two popular directions for extending VAEs to address these drawbacks.

### 1.1 Approximate posterior $q_\phi(\mathbf{z}|\mathbf{x})$:

To enhance the posterior expressiveness, Normalizing flow (NF) (Rezende & Mohamed, 2015) applies a sequence of invertible transformations to initial density $q_0(\mathbf{z})$ to achieve more expressive posteriors. $\beta$-VAE (Higgins et al., 2016) introduced a new parameter $\beta$ in balancing the reconstruction loss with disentangled latent representation $q_\phi(\mathbf{z}|\mathbf{x})$. Importance weighted Autoencoders (IWAE) (Burda et al., 2015) improved log-likelihood lower bound by importance weighting. It also approximated the posterior with multiple samples and enriched latent space representations. Saha et al. (2024) proposed the Aggregate Variational Autoencoder (AVAE), which estimates the posterior by Gaussian kernel density estimations (KDEs) to prevent issues like posterior collapse and improve the quality of the learned latent space.

### 1.2 Prior distribution $p(\mathbf{z})$ of latent variable $\mathbf{z}$:

Instead of approximating posterior, we can replace Gaussian prior with other flexible distributions. For instance, Dilokthanakul et al. (2016) used a simple Gaussian mixture prior and Tomczak & Welling (2018)

introduced a mixture prior called "VampPrior" which consists of a mixture distribution with components given by variational posteriors. The possibility of non-parametric priors is explored in Nalisnick & Smyth (2017), which utilized a truncated stick-breaking process. Hasnat et al. (2017) and Davidson et al. (2018) attempted to replace Gaussian prior by von Mises-Fisher(vMF) distribution. The intuition is that the Gaussian distribution may have limited coverage if the true latent space is hyperspherical. For data with heavy-tailed characteristics, Kim et al. (2024) replaced Gaussian prior with heavy-tailed distributions such as the Student's t-distribution, which allows the model to capture data with higher kurtosis, leading to more robust representations. Hao & Shafto (2023) utilized optimal transport theory design priors that enforce a coupling between the prior and data distribution.

Most variants of VAE along these two directions have the closed-form KL divergence. This is essential in implementation but somehow limits the potential applications of more flexible pairs of posterior and prior which has no closed-form of KL divergence but could better capture the latent data distribution and alleviate model collapse. To improve the flexibility of the choice of posterior and prior in VAE, in this paper, we estimate the posterior by the expectation of KDEs and derive a corresponding upper bound of KL-divergence, which has closed-form for many distributions. Such flexibility also makes the derivation of the optimal functional form of kernel possible. Note that Saha et al. (2024) directly employed a Gaussian kernel density estimator to estimate the posterior, which requires large number of samples to achieve decent performance. Our main contributions can be summarized as follows:

1. We first formulate the latent space learning process in VAE as a problem of kernel density estimation.

2. Inspired by the results in Bickel & Rosenblatt (1973) , we model the posterior in KL-divergence by KDE and derive an upper bound of KL-divergence, which has closed-form for many distributions. Some asymptotic results are established as well. See details in section 2.2.

3. After that, we utilize the conclusion from Bickel & Rosenblatt (1973) and show that the derived upper bound of KL-divergence is tightest when we employ Epanechnikov kernel. The derivation connects a quadratic functional with KL-divergence, which to our best knowledge, is the first attempt to bridge the concept of asymptotic distribution of KDEs with VAE.

4. Thanks to the reparametrization trick, the implementation of Epanechnikov kernel in VAE is straightforward and doesn't require too many samples from latent space. We conduct a detailed comparison of the performance of Epanechnikov VAE (EVAE) and Gaussian VAE in many benchmark image datasets under standard encoder-decoder structure. The experiments not only demonstrate the superiority of EVAE over VAE and other baselines in the quality of reconstructed images, as measured by the FID score and Sharpness(Tolstikhin et al., 2018) but also illustrate that EVAE has reasonable time efficiency compared to VAE.

The remaining sections of this paper are organized as following schema. In section 2, we provide some preliminaries involving VAE, kernel density estimation and few assumptions. In section 3, we show that the optimal functional form of kernel in bounding the KL-divergence is Epanechnikov kernel. Based on results in section 3, we propose the Epanechnikov VAE (EVAE) in section 4. The comparisons between EVAE, vanilla VAE and other baselines in benchmark datasets are illustrated in section 5. Section 6 discusses few characteristics and limitations of EVAE and suggests some future directions.

## 2 Preliminary

### 2.1 VAE formulation

Consider a dataset $\mathbf{X} = \{\mathbf{x}^{(i)}\}_{i=1}^n$ consists of $n$ i.i.d samples from space $\mathcal{X}$ whose dimension is $d$. In VAE, we assume that every observed data $\mathbf{x}^{(i)} = (x_1^{(i)}, x_2^{(i)}, ..., x_d^{(i)}) \in \mathcal{X}$ is generated by a latent variable $\mathbf{z}^{(i)} \in \mathcal{Z}$ whose dimension is $p$. Then the data generation process can be summarized in 2 steps. It first produces a variable $\mathbf{z}^{(i)}$ from some prior distributions $p(\mathbf{z})$. Then given the value of $\mathbf{z}^{(i)}$, an observed value $\mathbf{x}^{(i)}$ is generated from certain conditional distribution $p_\theta(\mathbf{x}|\mathbf{z})$. We typically assume $d > p$ and likelihood $p_\theta(\mathbf{x}|\mathbf{z})$ are differentiable distributions w.r.t $\theta$ and $\mathbf{z}$.

To maximize the empirical log likelihood $\log p_\theta(\mathbf{x})$, we need to evaluate the intractable integral in the form

$$\log p_\theta(\mathbf{x}) = \log \int p_\theta(\mathbf{x}, \mathbf{z})d\mathbf{z} = \log \int p_\theta(\mathbf{x}|\mathbf{z})p(\mathbf{z})d\mathbf{z},$$

which is not available in most cases. Fortunately, we can instead maximizing its log evidence lower bound (ELBO) $\mathcal{L}$ with the help of Jensen's inequality:

$$\log p_\theta(\mathbf{x}) \geq \mathop{\mathbb{E}}_{\mathbf{z} \sim q_\phi(\cdot|\mathbf{x})} [\log p_\theta(\mathbf{x}|\mathbf{z})] - KL(q_\phi(\mathbf{z}|\mathbf{x})||p(\mathbf{z})), \tag{1}$$

where the RHS of inequality (1) is called evidence lower bound (ELBO) and $KL(\cdot||\cdot)$ represents the KL-divergence between two distributions.

The conditional likelihood $p_\theta(\mathbf{x}|\mathbf{z})$, approximate posterior $q_\phi(\mathbf{z}|\mathbf{x})$ and the prior distribution $p(\mathbf{z})$ can be chosen independently. For convenience, most applications of VAE employ Gaussian parametrization for all three likelihoods. Since we would like to investigate optimal posterior and prior rather than the form of conditional likelihood $p_\theta(\mathbf{x}|\mathbf{z})$, we can assume a multivariate Bernoulli or Gaussian model w.r.t $p_\theta(\mathbf{x}|\mathbf{z})$ for simplicity. For example, under multivariate Gaussian, we have $\log p_\theta(\mathbf{x}|\mathbf{z}) = \log \mathcal{N}(\mathbf{x}; g_\theta(\mathbf{z}), I)$. As to multivariate Bernoulli model, we have $\log p_\theta(\mathbf{x}|\mathbf{z}) = \sum_{i=1}^{d}[x_i \log (g_\theta(\mathbf{z})_i) + (1 - x_i)\log (1 - g_\theta(\mathbf{z})_i)]$ where $g_\theta(\mathbf{z}) : \mathcal{Z} \to \mathcal{X}$ are typically neural-network parametrizations.

To maximize ELBO, we now need to minimize the following target function for given data $\mathbf{x}$:

$$\mathop{\mathbb{E}}_{\mathbf{z} \sim q_\phi(\cdot|\mathbf{x})} [-\log p_\theta(\mathbf{x}|\mathbf{z})] + KL(q_\phi(\mathbf{z}|\mathbf{x})||p(\mathbf{z})). \tag{2}$$

The two terms in equation (2) are named as "reconstruction error" and "divergence" or "regularization term", respectively. The "divergence" term regularizes the mismatch between approximate posterior and prior distribution.

## 2.2 Kernel density estimation

### 2.2.1 Model the posterior as the expectation of kernel density estimator

A common assumption of latent space in VAE is the factorization of approximate posterior, i.e. dimensions/features of latent space are independent with each other. Under this assumption, we only need to consider the formulation of KDE of posterior $q_\phi(\mathbf{z}|\mathbf{x})$ and prior $p(\mathbf{z})$ in one-dimensional case. Similar arguments can be applied to multi-dimensional cases by additivity of KL-divergence under independence. For the consistency of notations, we still use bold letter $\mathbf{x}$ and $\mathbf{z}$ to denote $x$ and $z$ in one-dimensional KDE.

Given $Y_1, ..., Y_m$ be i.i.d random variables with a continuous density function $f$, Parzen (1962) and Rosenblatt (1956) proposed kernel density estimate $f_n(y)$ for estimating $f(y)$ at a fixed point $y \in \mathbb{R}$:

$$f_m(y) = \frac{1}{mb(m)} \sum_{i=1}^{m} K\left[\frac{y - Y_i}{b(m)}\right] = \frac{1}{b(m)} \int K\left[\frac{y - t}{b(m)}\right] dF_m(t), \tag{3}$$

where $F_m$ is the sample distribution function, $K$ is an appropriate kernel function such that $\int K(y)dy = 1$ and the positive number $b_m$, which typically relies on the sample size $m$, is called bandwidth such that $b(m) \to 0, mb(m) \to \infty$ as $m \to \infty$[1].

On the other hand, with inequality $\log (t) \leq t - 1$ (for $t > 0$) we can bound KL-divergence as follows:

$$KL(q||p) = \int q(\mathbf{z})\log \frac{q(\mathbf{z})}{p(\mathbf{z})}d\mathbf{z} \leq \int q(\mathbf{z})(\frac{q(\mathbf{z})}{p(\mathbf{z})} - 1)d\mathbf{z} = \int \frac{(q(\mathbf{z}) - p(\mathbf{z}))^2}{p(\mathbf{z})}d\mathbf{z}. \tag{4}$$

---

[1]We omit the limits of integrals if they are $-\infty$ to $\infty$.

Let the posterior $q_\phi(\mathbf{z}|\mathbf{x}) = \mathbb{E}_0(q_{m,\phi}(\mathbf{z}))$, where

$$q_{m,\phi}(\mathbf{z}) = \frac{1}{mb(m)} \sum_{j=1}^m K_{\phi,\mathbf{x}} \left[ \frac{\mathbf{z} - \mathbf{Z}_j}{b(m)} \right], \quad \mathbf{Z}_j \overset{i.i.d}{\sim} p(\tilde{\mathbf{z}}).$$

is a kernel density estimator of $q_\phi(\mathbf{z}|\mathbf{x})$ with kernel $K_{\phi,\mathbf{x}}$, given KDE sample size $m$, data point $\mathbf{x}$ and parameter $\phi$. Expectation $\mathbb{E}_0$ is taken w.r.t the samples from prior distribution $p(\tilde{\mathbf{z}})$. In other words, we model the posterior as the expectation of the kernel density estimator. The subscript $\phi$ of kernel function implies that the parameters in kernel can be learned by neural networks.

By inequality (4) and Jensen inequality, we obtain

$$KL(q_\phi(\mathbf{z}|\mathbf{x})||p(\mathbf{z})) \leq \int \frac{[\mathbb{E}_0(q_{m,\phi}(\mathbf{z})) - p(\mathbf{z})]^2}{p(\mathbf{z})} d\mathbf{z} \leq \mathbb{E}_0 \left[ \int \frac{(q_{m,\phi}(\mathbf{z}) - p(\mathbf{z}))^2}{p(\mathbf{z})} d\mathbf{z} \right]. \tag{5}$$

Our main goal is to find the optimal kernel function $K_{\phi,\mathbf{x}}$ with other model parameters $\phi$ and data point $\mathbf{x}$ fixed. To this end, we obtain a new lower bound of log-likelihood of data:

$$\begin{aligned} \log p_\theta(\mathbf{x}) &\geq \mathbb{E}_{q_\phi(\mathbf{z}|\mathbf{x})}[\log p_\theta(\mathbf{x}|\mathbf{z})] - KL(q_\phi(\mathbf{z}|\mathbf{x})||p(\mathbf{z})) \\ &\geq \mathbb{E}_{q_\phi(\mathbf{z}|\mathbf{x})}[\log p_\theta(\mathbf{x}|\mathbf{z})] - \mathbb{E}_0 \left[ \int \frac{(q_{m,\phi}(\mathbf{z}) - p(\mathbf{z}))^2}{p(\mathbf{z})} d\mathbf{z} \right]. \end{aligned} \tag{6}$$

Since we model the posterior as the expectation of the kernel density estimator, the number of latent variable $Z$ generated from prior distribution, which is $m$, is just used for theoretical derivation and we don't need to sample too many number of latent variable $Z$ in practice.

In the kernel density estimation theory, Epanechnikov (1969) showed that the Epanechnikov kernel minimizes $\int \mathbb{E}_0[q_{m,\phi}(\mathbf{z}) - p(\mathbf{z})]^2 d\mathbf{z}$ asymptotically, which gives us a hint of the potential optimization of kernel $K$ in right hand side of inequality (5). Before we derive the optimal functional form of kernel, we 'd like to briefly introduce some main assumptions and notations in the following section.

### 2.2.2 Assumptions

We mainly borrow the terminologies and assumptions in Rosenblatt (1956) which is the pioneering work in measuring deviations of density function estimates. Note that Rosenblatt (1956) studied the asymptotic distribution of the quadratic functional $\int [f_n(t) - f(t)]^2 a(t) dt$ under appropriate weight function $a$ and conditions as sampling size $n$ approaches to infinity. In inequality (5), we just saw that the weight function is the reciprical of prior density in our case. Assumptions $\mathbf{A1} - \mathbf{A4}$ are listed as follows:

(**A1**): The kernel function $K$ is bounded, integrable, symmetric (about 0) and $\int K(t)dt = 1, \int t^2 K(t)dt < \infty, \int K^2(t)dt < \infty$. Also, $K$ either (a) is supported on an closed and bounded interval $[-B, B]$ and is absolutely continuous on $[-B, B]$ with derivative $K'$ or (b) is absolutely continuous on the whole real line with derivative $K'$ satisfying $\int |K'(t)|^k dt < \infty, k = 1, 2$. Moreover,

$$\int_{t \geq 3} |t|^{\frac{3}{2}} [\log(\log|t|)]^{\frac{1}{2}} [|K'(t)| + |K(t)|] dt < \infty$$

(**A2**): The underlying density $f$ is continuous, positive and bounded.

(**A3**): Squared density $f^{1/2}$ is absolutely continuous and its derivative is bounded in absolute value.

(**A4**): The second derivative $f''$ exists and is bounded.

We can see that the Gaussian kernel satisfies those assumptions, indicating that the kernel density estimation theory also works for Gaussian VAE. Similarly, most priors and posteriors we listed in section 1 lie in conditions $\mathbf{A1} - \mathbf{A4}$, demonstrating the promising applications of kernel estimate theory in variants of VAE.

## 3   Choice of kernel

Let $f_m(t)$ be a kernel density estimate of a continuous density function $f$ at $t$, as defined in equation (3), we construct a statistic $T_m$ as follows:

$$T_m = mb(m) \int [f_m(t) - f(t)]^2 a(t)dt\,,$$

where $a(t)$ is an appropriate weight function. We now restate the main result of Bickel & Rosenblatt (1973) as Theorem 3.1:

**Theorem 3.1 (Bickel & Rosenblatt (1973))** *Let $A1 - A4$ hold and suppose that the weight function $a$ is integrable piecewise continuous and bounded. Suppose $b(n) = o(n^{-\frac{2}{9}})$ and $o(b(n)) = n^{-\frac{1}{4}}(log(n))^{\frac{1}{2}}(loglogn)^{\frac{1}{4}}$ as $n \to \infty$, then $b^{-\frac{1}{2}}(n)(T_n - I(K)\int f(t)a(t)dt)$ is asymptotically normally distributed with mean 0 and variance $2J(K)\int a^2(t)f^2 dt$ as $n \to \infty$, where*

$$I(K) = \int K^2(t)dt, \quad J(K) = \int \left[ \int K(t+y)K(t)dt \right]^2 dy\,. \tag{7}$$

In other words, under Theorem 3.1, we have

$$\mathbb{E}[T_m] \to I(K) \int f(t)a(t)dt \quad , \text{ as } m \to \infty\,.$$

Let $mb(m)/n = 1$ and $b(m)$ satisfies the conditions in Theorem 3.1, by the asymptotic result of Theorem 3.1, the inequality (5) becomes[2]:

$$
\begin{aligned}
KL(q_\phi(\mathbf{z}|\mathbf{x})||p(\mathbf{z})) &\leq \frac{mb(m)}{n}\mathbb{E}_0 \left[ \int \frac{(q_{m,\phi}(\mathbf{z}) - p(\mathbf{z}))^2}{p(z)}d\mathbf{z} \right] \\
&= \frac{1}{n}\mathbb{E}_0 \left[ mb(m) \int \frac{(q_{m,\phi}(\mathbf{z}) - p(\mathbf{z}))^2}{p(\mathbf{z})}d\mathbf{z} \right] \\
&\stackrel{\text{n large}}{\approx} \frac{I(K_{\phi,\mathbf{x}})\int p(\mathbf{z})\frac{1}{p(\mathbf{z})}d\mathbf{z}}{n} = B\frac{I(K_{\phi,\mathbf{x}})}{n}\,,
\end{aligned}
\tag{8}
$$

where B is the length of support interval of prior $p(\mathbf{z})$. For simplicity, we assume all data points have the same length of support interval. Note that if the prior has infinite length of support, the inequality (8) becomes theoretically useless. However, in practice, we found that parameter B is still useful for Gaussian prior. See section 5.4 and Appendix E. Instead of finding the optimal functional form of posterior and prior in the KL-divergence, we can now find the kernel which gives the tightest upper bound of KL-divergence i.e. the kernel $K_{\phi,\mathbf{x}}^*$ minimizing $I(K_{\phi,\mathbf{x}})$ given fixed parameters and data point $\mathbf{x}$. Lemma 3.2 shows that Epanechnikov kernel is the optimal choice.

**Lemma 3.2** *Let $\mathcal{K}$ be the set of all $L^1(-\infty, \infty)$ nonnegative functions $K$ satisfying*

$$\int K(t)dt = 1, \int tK(t)dt = \mu, \int (t-\mu)^2 K(t)dt = \frac{1}{5}r^2$$

*where $\mu \in (-\infty, \infty), r > 0$. Then the functional $I(K)$ in equation (7) is minimized on $\mathcal{K}$ uniquely by*

$$
K^*(t) = \begin{cases} \frac{3}{4r}\left(1 - \left(\frac{t-\mu}{r}\right)^2\right) & t \in [\mu - r, \mu + r] \\ 0 & otherwise \end{cases}
$$

*and $min_K I(K) = I(K^*) = \frac{3}{5r}$. The optimal kernel $K^*$ is named as Epanechnikov kernel (Epanechnikov, 1969).*

---

[2]For the consistency of notations, we still use bold $\mathbf{x}$ and $\mathbf{z}$ to denote one-dimensional variable $x$ and $z$ in inequality (equation 8) under the context of Theorem 3.1.

The proof is based on the idea of Lagrange multiplier. Few observations from Lemma 3.2 and Figure S4 in Appendix A.1 (plots of standard Epanechnikov kernel ($\mu = 0, r = 1$) and standard Gaussian):

1. The support of Epanechnikov kernel is closed and bounded (i.e compact). However, the support of Gaussian distribution is unbounded, which may lead to the noisy or blurry regenerated images. Thus we posit that Epanechnikov kernel could regenerate sharper regenerated. Empirical evidence is reported in section 5.

2. The distribution function of Epanechnikov kernel lies in the "location-scale" family as well, which facilitates the implementation of Epanechnikov kernel VAE by reparametrization tricks. See section 4 for details.

3. $I(K^*)$ doesn't include $\mu$, indicating that $\mu$ is only optimized in the reconstruction term in ELBO. The parameter B in equation (8) controls the support of prior distribution. If we set B as a constant all the time, then EVAE can be connected to $\beta$-VAE (Higgins et al., 2016) which adds a weight parameter $\beta$ in front of the KL term.

## 4 Epanechnikov VAE

Getting inspired by the functional optimality of Epanechnikov kernel in controlling KL-divergence, we propose the Epanechnikov Variational Autoencoder (EVAE) whose resampling step is based on Epanechnikov kernel. There are two main differences between EVAE and VAE. The latent distribution in EVAE is assumed to be estimated by the Epanechnikov kernel rather than multivariate isotropic Gaussian. And EVAE is trained to minimize a different target function (9):

$$\mathbb{E}_{z \sim q_{\phi}(\cdot|\mathbf{x})} \left[-\log p_\theta(\mathbf{x}|\mathbf{z})\right] + B \frac{I(K^*_{\phi,\mathbf{x}})}{n} , \tag{9}$$

where $n$ can be the sample size or minibatch size, $\phi$ are outputs of the encoding network, $K^*_{\phi,\mathbf{x}}$ is Epanechnikov kernel with trainable neural network parameter $\phi$ and data point $\mathbf{x}$, and support parameter $B$ is a constant. Note that target function (9) is an upper bound of equation (2) used in ordinary VAE. Suppose we have $M$ data points in each minibatch, the sample version of equation (9) at $i$-th data point $\mathbf{x}^{(i)}$ is

$$\tilde{\mathcal{L}}(\theta, \phi, \mathbf{B}; \mathbf{x}^{(i)}) \approx -\frac{1}{L} \sum_{l=1}^{L} (\log p_\theta(\mathbf{x}^{(i)})|\mathbf{z}^{(i,l)}) + \frac{3B}{5M} \sum_{k=1}^{p} \frac{1}{r_k^{(i)}} , \tag{10}$$

where $i \in \{1, 2, ..., M\}$, p is the dimension for latent space, B is the support length of prior (which we set to be constant for each hidden dimension and all data points), $\mu^{(i)} = (\mu_1^{(i)}, ..., \mu_p^{(i)}), \mathbf{r}^{(i)} = (r_1^{(i)}, ..., r_p^{(i)})^3$ are outputs of the encoding networks with variational parameters $\phi$ and $M$ is minibatch size. In vanilla VAE, Kingma & Welling (2013) suggested that the number of regenerated samples $L$ can be set to 1 as long as the minibatch size is relatively large, which is also the case for EVAE. To sample latent variables from the posterior, we need to derive the density of $q_\phi(\mathbf{z}|\mathbf{x}^{(i)})$:

$$
\begin{aligned}
q_\phi(\mathbf{z}|\mathbf{x}^{(i)}) &= \mathbb{E}_0 \left[ \frac{1}{mb(m)} \sum_{j=1}^{m} K^*_{\phi,\mathbf{x}^{(i)}} \left( \frac{\mathbf{z} - \mathbf{Z}_j}{b(m)} \right) \right] \\
&= \mathbb{E}_0 \left[ \frac{1}{b(m)} K^*_{\phi,\mathbf{x}^{(i)}} \left( \frac{\mathbf{z} - \mathbf{Z}_1}{b(m)} \right) \right] \\
&= \int \frac{1}{b(m)} K^*_{\phi,\mathbf{x}^{(i)}} \left( \frac{\mathbf{z} - \tilde{\mathbf{z}}}{b(m)} \right) p_{\mathbf{z}}(\tilde{\mathbf{z}}) d\tilde{\mathbf{z}} ,
\end{aligned}
\tag{11}
$$

---

[3] The encode network output $\mu^{(i)}$ is used to sample from Epanechnikov based posterior and only optimized in the reconstruction term.

where $p_{\mathbf{z}}$ is the density of prior distribution. Given a data point $\mathbf{x}^{(i)}$, let $q_\phi(\mathbf{z}|\mathbf{x}^{(i)})$ be the density of random variable $Z(\mathbf{x}^{(i)})$, $K^*_{\phi,\mathbf{x}^{(i)}}$ be the density of random variable $K^*(\mathbf{x}^{(i)})$ and prior $p_{\mathbf{z}}$ is density of the random variable $Z$, then it's clear to see that the posterior is now a convolution between two random variables, i.e.

$$Z(\mathbf{x}^{(i)}) = b(m)K^*(\mathbf{x}^{(i)}) + Z\,. \tag{12}$$

Essentially, identity (12) decomposes the posterior into two parts. One is the prior information, the other represents the incremental updates from new information induced by data $\mathbf{x}^{(i)}$. The Epanechnikov kernel can be viewed as the "optimal" direction of perturbing the prior distribution and the coefficient $b(m)$ can be interpreted as "step size", which controls the deviation of posterior from the prior. The resampling step in EVAE can be divided into two parts as well, as described in Algorithm 1 where we used uniform distribution as the prior in EVAE. For the sake of finite support and simplicity, we assume that the prior distribution is uniformly distributed and step size $b(m)$ is the same for all dimensions. The theoretical conditions for $b(m)$ is demanding. In practice we found that setting step size as $b(100) = 100^{-2/9} \approx 0.3594$ is good enough.

---

**Algorithm 1** Resampling step in a minibatch of EVAE

---

**Require:** Latent space dimension $d_z$; Minibatch size $M$. Step size $b(m)$. Prior support interval length $B$. Mean $\mu_\phi(\mathbf{x})$, spread $\mathbf{r}_\phi(\mathbf{x})$ are all learned by encoder networks given input $\mathbf{x}$ and have dimension $M \times d_z$.
1: Sample a $M \times d_z$ matrix $\mathbf{U}$ where each $(i, j)$ entry of the random matrix $\mathbf{U}$ is sampled from $\mathrm{Unif}[-B/2, B/2]$.
2: Sample a $M \times d_z$ matrix $\mathbf{K}$ where each $(i, j)$ entry of the random matrix $\mathbf{K}$ is sampled from an standard Epanechnikov kernel supported on $[-1, 1]$.
3: Shift and scale sampled $\mathbf{K}$ by the location-scale formula: $\mathbf{Z} = \mu_\phi(\mathbf{x}) + \mathbf{r}_\phi(\mathbf{x}) \odot \mathbf{K}$.
4: **Return:** $b(m) \odot \mathbf{Z} + \mathbf{U}$

---

---

**Algorithm 2** Sampling from centered Epanechnikov kernel supported on $[-1, 1]$

---

1: Sample $U_1, U_2, U_3 \overset{i.i.d}{\sim} \mathrm{Unif}[-1, 1]$.
2: Set $U = \mathrm{Median}(U_1, U_2, U_3)$
3: **Return:** $U$

---

In Algorithm 1, we apply reparametrization trick to sample $\mathbf{z}$ from general Epanechnikov kernel, i.e $\mathbf{z}^{(i,l)} = \mu^{(i)} + \mathbf{r}^{(i)} \odot \mathbf{k}^{(l)}$, where $\mathbf{k}^{(l)}$ is sampled from standard Epanechnikov kernel supported on $[-1, 1]$ as it lies in the "location-scale" family. We use $\odot$ to signify the element-wise product. There are many ways to sample from standard Epanechnikov kernel, such as accept-rejection method. For efficiency, we provide a faster sampling procedure in Algorithm 2. The theoretical support is given in Section A.2.

As we mentioned in section 3, the regularization term in equation (9) doesn't include $\mu_\phi(\mathbf{x})$. It is only optimized in the reconstruction term in equation (10). In Gaussian VAE, the prior distribution has infinite support which limits the capability of latent space while the extra parameter B induced by KDEs controls the support of prior distribution, which makes EVAE more flexible. To maximize target function empirically, the spread parameter $\mathbf{r}$ tends to be small, which may lead to numerical issue in back propagation. However, the numerator B in the second part of equation (10) plays an rule to stabilize scaling effect.

## 5 Experiments

In this section, we will compare the proposed EVAE model with (vanilla) VAE whose posterior and prior are modelled by isotropic multivariate Gaussian in real datasets. To assess the quality of reconstructed images, we employ the Frechet Inception Distance (FID) score (Heusel et al., 2017) to measure the distribution of generated images with the distribution real images. The lower, the better. Inception_v3 (Szegedy et al., 2016) is employed as default model to generate features of input images, which is a standard implementation in generative models. In section 3, we mentioned that the compact support for Epanechnikov kernel could help EVAE generate less blurry or noisy images. To check this claim empirically, we calculated sharpness

(Tolstikhin et al., 2018) for generated images by a $3 \times 3$ Laplace filter. See details in Appendix C. The pytorch codes for the implementation of EVAE and experiments are available in supplementary materials. All simulations and experiments are performed on a laptop with 12th Gen Intel(R) Core(TM) i7-12700H (2.30 GHz) ,16.0 GB RAM and NVIDIA 4070 GPU.

## 5.1 Benchmark datasets

We trained EVAE and VAE on four benchmark datasets: MNIST (Deng, 2012), Fashion-MNIST (Xiao et al., 2017), CIFAR-10 (Krizhevsky, 2009) and CelebA (Liu et al., 2015). The detailed information for training parameters are attached in Appendix C. For comparison, we applied a classical CNN architecture (Appendix B) in encoding and decoding part for all datasets. Consequently, the main differences between EVAE and VAE in implementation stem from the resampling step and target function in training procedure. All FID scores and sharpness are based on hold-out samples. We also evaluated VAE and EVAE with different dimensions $d_z$ of latent spaces. Table 1 and Table 2 summarize the results on four datasets with uniform prior support parameter B=0.1 in EVAE. In terms of FID score, we observe that EVAE has an edge in high dimensions $(d_z = 32, 64)$, indicating the better quality of reconstructed images from EVAE. Additionally, when $d_z = 64$, EVAE generate higher sharpness of reconstructed images for all datasets, as illustrated in Table 1 and Table 2. This result empirically justifies the positive effect of having compact support in posteriors and priors. As to CIFAR-10, EVAE has larger sharpness in high dimensions $(d_z = 32, 64)$ while it is relatively mediocre in low dimensions. Possible reasons include low-resolution(blurriness) of original images and deficient expressibility of simple CNN models in low dimensions. But the validation reconstruction loss curves in Appendix F indeed authenticate the superiority of EVAE over benchmark datasets, including CelebA.

Table 1: VAE and EVAE results on MNIST and Fashion-MNIST datasets

| Datasets | MNIST | | | | Fashion-MNIST | | | |
|---|---|---|---|---|---|---|---|---|
| | VAE | | EVAE | | VAE | | EVAE | |
| $d_z$ | FID | Sharpness | FID | Sharpness | FID | Sharpness | FID | Sharpness |
| 8 | 15.70 | 0.0211 | 14.74 | 0.0243 | 41.47 | 0.0157 | 36.18 | 0.0176 |
| 16 | 11.93 | 0.0252 | 10.49 | 0.0305 | 38.85 | 0.0159 | 26.71 | 0.0216 |
| 32 | 11.97 | 0.0249 | 7.92 | 0.0338 | 38.74 | 0.0163 | 18.47 | 0.0251 |
| 64 | 11.97 | 0.0246 | 5.13 | 0.0360 | 39.31 | 0.0161 | 12.02 | 0.0267 |

Table 2: VAE and EVAE results on CIFAR10 and CelebA datasets

| Datasets | CIFAR-10 | | | | CelebA | | | |
|---|---|---|---|---|---|---|---|---|
| | VAE | | EVAE | | VAE | | EVAE | |
| $d_z$ | FID | Sharpness | FID | Sharpness | FID | Sharpness | FID | Sharpness |
| 8 | 230.14 | 0.0353 | 226.61 | 0.0337 | 198.71 | 0.0147 | 198.74 | 0.0154 |
| 16 | 181.39 | 0.0350 | 164.20 | 0.0355 | 152.45 | 0.0148 | 149.73 | 0.0154 |
| 32 | 149.71 | 0.0348 | 123.47 | 0.0359 | 110.05 | 0.0156 | 97.54 | 0.0162 |
| 64 | 144.66 | 0.0355 | 79.65 | 0.0369 | 77.24 | 0.0163 | 65.33 | 0.0167 |

## 5.2 Effect of uniform prior support parameter B

Table 3 summarizes the performance of EVAE with uniform prior under different values of B in different datasets. ($d = 64$ and other settings are the same with original experiments). We can see EVAE becomes worse when B is relatively large. One reason maybe that prior distribution with larger support is likely to generate outlier samples and lower down the quality of reconstructed sample. Additionally, the large value of B will add more weight on the regularization term of the new target function (10) of EVAE, which trades off the reconstruction performance of EVAE.

Table 3: The **FID** score of EVAE with different constant value of B in MNIST, Fashion-MNIST and CIFAR-10 datasets.($d_z = 64$ in all cases)

| Support parameter B | MINST | Fashion-MNIST | CIFAR-10 |
|---|---|---|---|
| B = 0.01 | 4.96 | 12.2 | 81.1 |
| B = 0.1 | 5.14 | 12.4 | 79.4 |
| B = 1 | 7.13 | 16.9 | 93.9 |
| B = 10 | 11.29 | 33.7 | 153.3 |
| B = 20 | 12.93 | 43.1 | 168.7 |

## 5.3 Reconstruction samples

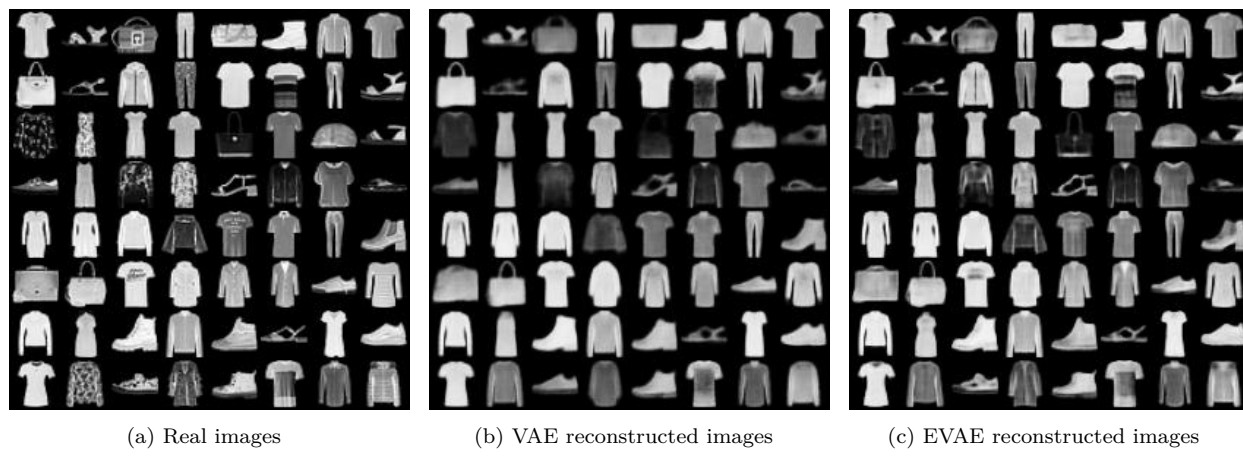

(a) Real images          (b) VAE reconstructed images          (c) EVAE reconstructed images

Figure 1: (a) Sampled real images from hold-out samples in Fashion-MNIST (b) Reconstructed images by VAE. (c) Reconstructed images by EVAE. Dimension $d_z = 64$ for both models.

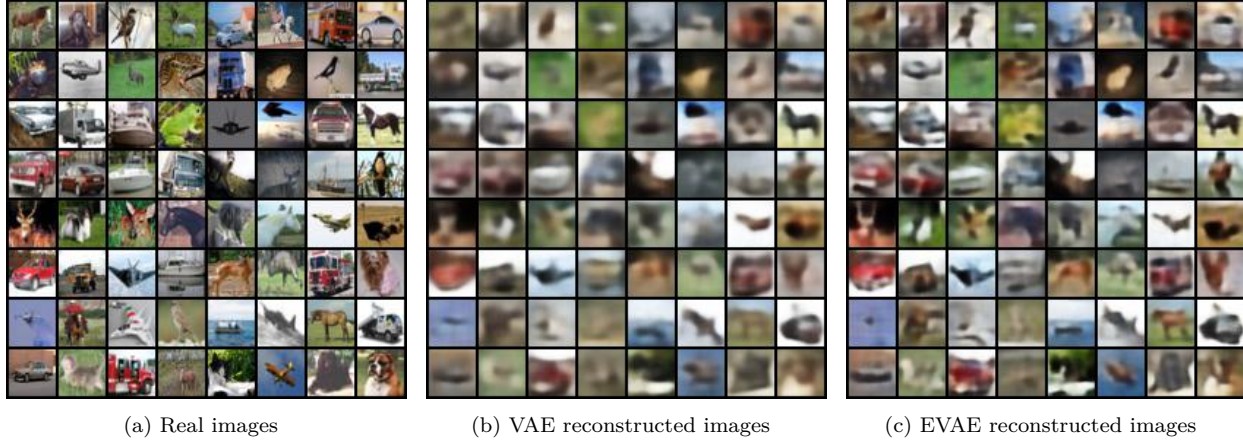

(a) Real images          (b) VAE reconstructed images          (c) EVAE reconstructed images

Figure 2: (a) Sampled real images from hold-out samples in CIFAR-10 (b) Reconstructed images by VAE. (c) Reconstructed images by EVAE. Dimension $d_z = 64$ for both models.

Figure 1 and Figure 2 present some test reconstructed samples from trained VAE and EVAE with $d_z = 64$ of Fashion-MNIST and CIFAR10, respectively. We can see many images generated from EVAE are able to pick up local features better than the VAE. And reconstructed samples from CIFAR-10 are clearer and closer to the original images, as indicated by large gap between FID scores. See more examples of MINST and

CelebA-64 datasets in Appendix D. The statistical analysis from binomial test based on total experiments (Appendix C) shows that EVAE significantly outperforms VAE in FID and sharpness.

## 5.4 EVAE with Gaussian prior

To check whether the superior performance of EVAE over Gaussian VAE comes from the new prior (uniform) or new posterior (convolution between prior and KDE ), we performed extra experiments for EVAE with Gaussian prior. Table 6 and Table 7 in Appendix E compare the FID score and sharpness of EVAE (with Gaussian prior and B=0.1) with VAE and results show that EVAE still outperforms VAE. To this point, we can say the superior performance of EVAE mainly comes from the introduction of KDE based posterior.

## 5.5 Unconditional samples

Figure 3 demonstrates the unconditional samples from MINST dataset with EVAE [4] and VAE and different values of parameter B. In other words, we directly sample from prior $p(z)$, multiply it by B and feed latent samples into decoder to obtain the unconditional samples. We observe that the larger the value of B is, the more diversed novel samples will be generated from EVAE. This makes sense as smaller value of B will make the model focus more on the perturbation part (KDE based) of identity (12), leading to the prior collapse. To this point, we can view the parameter B as a way to trade off the reconstruction and sample diversity.

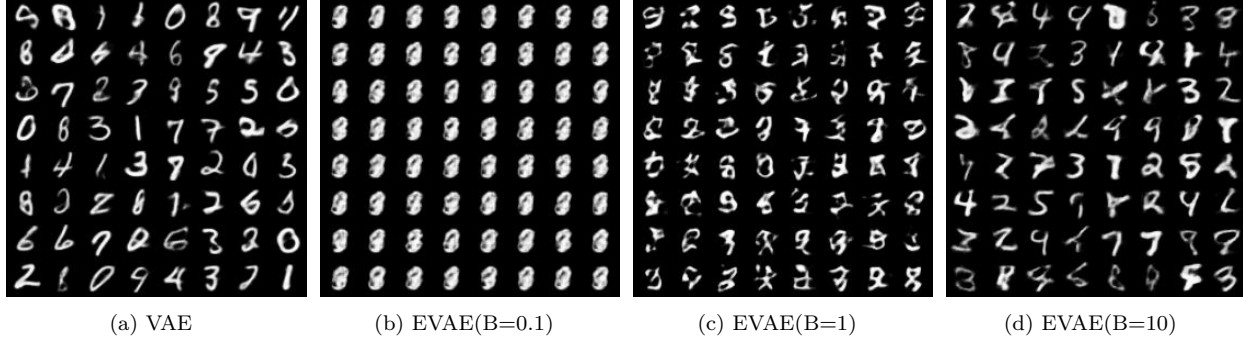

|  (a) VAE  |  (b) EVAE(B=0.1)  |  (c) EVAE(B=1)  |  (d) EVAE(B=10)  |

Figure 3: (a) Unconditional samples generated from VAE (MNIST dataset) (b) Unconditional samples generated from EVAE with B=0.1 (c) Unconditional samples generated from EVAE with B=1 (d) Unconditional samples generated from EVAE with B=10

## 5.6 Comparisons with baselines

In this section we conducted extra experiments comparing EVAE with open-source techniques that improve the representational capacity of the encoder such as $\beta-$VAE (Higgins et al., 2016), IWAE (Burda et al., 2015), WAE (Tolstikhin et al., 2018) and HVAE (Davidson et al., 2018) on three datasets. We employed uniform prior in EVAE and we ran all experiments for three times. We also reported mean and standard deviation of FID score (for testing set reconstruction quality). Same for sharpness. For fair comparisons, we used same encoder and decoder CNN architecture for all baselines. The only difference come from posterior modeling and prior modeling. All latent dimension $d_z$ are set to be 64. Other hyperparameters and training details are the same with details in Appendix B and C. From Table 4, we see that EVAE with $B = 0.1$ performs universally good in FID and sharpness. Our support length parameter $B$ plays the similar role with $\beta$ in $\beta-$VAE, which verifies the intuition in the discussion section. In other word, $B$ and $\beta$ somehow control the reconstruction quality and sample diversity. For other baselines, WAE(MMD) performs also well in many cases, which is expected as it generalizes the KL divergence to the Wasserstein distance. IWAE also has decent performance by utilizing importance weight sampling to shrink the gap between ELBO and log-likelihood, where $k = 5$ represents the number of samples in the importance sampling step. Hyperspherical

---

[4]We employed uniform prior in the rest of experiments. Section 5.4 already showed that there is no big difference for the model performance between uniform prior and Gaussian prior. The training details follows Appendix B and C.

VAE(HVAE) doesn't perform well in our experiments, which might be due to the complexity of von-Mises Fisher prior and choice of dimension of hypershere manifold. All the methods shared similar running time.

Table 4: EVAE and other baselines

| Models | MNIST | | Fashion-MNIST | | CIFAR10 | |
|---|---|---|---|---|---|---|
| | FID | Sharpness | FID | Sharpness | FID | Sharpness |
| VAE | 11.9(0.1) | 0.025(1.5E-4) | 39.1(0.3) | 0.016(3E-4) | 142.9(1.7) | 0.035(7E-4) |
| EVAE($B=0.1$) | **4.9(0.2)** | **0.036(2.1E-4)** | **12.2(0.2)** | **0.027(2.1E-4)** | **79.7(0.1)** | **0.036(7E-4)** |
| EVAE($B=10$) | 11.0(0.5) | 0.03(5E-5) | 32.6(0.5) | 0.019(4E-4) | 154.8(2.6) | 0.035(6E-4) |
| $\beta$-VAE($\beta=0.1$) | 7.8(0.2) | 0.033(4E-4) | 19.5(0.5) | 0.023(5E-4) | 84.4(0.8) | 0.035(5E-4) |
| $\beta$-VAE($\beta=5$) | 28.2(0.5) | 0.013(1E-4) | 74.5(2.0) | 0.01(2E-4) | 239.8(1.9) | 0.034(1.4E-3) |
| IWAE($k=5$) | 10.5(0.7) | 0.03(5E-5) | 26.8(0.1) | 0.02(3E-4) | 125.4(1.6) | 0.036(7E-4) |
| WAE(MMD) | 8.9(0.2) | 0.03(5E-5) | 22.1(0.5) | 0.021(6E-4) | 140.1(7.2) | 0.034(7E-4) |
| HVAE | 11.5(0.2) | 0.025(3E-4) | 36.8(0.3) | 0.016(1E-4) | 195.2(0.8) | 0.035(1E-3) |

### 5.7 Time efficiency of EVAE

One advantage of kernel posterior proposed in equation (11) is that the quadratic functional $I(K)$ has a closed form for many distributions, while the closed form of KL divergence in standard ELBO can be hard to derive when we want to use complicated posterior and prior. For example, the KL divergence becomes piecewise for the uniform prior and posterior. In those cases, time-consuming Monte Carlo simulations might be needed.

To explore the time efficiency of EVAE, we performed additional experiments in Table 5 to compare the average and standard error of the training time (in seconds) per epoch (10 epochs total for each experiment) for VAE and EVAE in CIFAR10 and CelbeA-64.

Table 5: Training time of EVAE and VAE per epoch in seconds

| Latent space dimension $d_z$ | CIFAR10 | | CelebA-64 | |
|---|---|---|---|---|
| | VAE(std) | EVAE(std) | VAE(std) | EVAE(std) |
| $d_z = 8$ | 7.21(0.28) | 7.9(0.19) | 208.94(5.95) | 208.42(4.33) |
| $d_z = 16$ | 7.22(0.43) | 7.99(0.19) | 209.10(5.94) | 210.88(5.35) |
| $d_z = 32$ | 7.9(0.6) | 8.87(0.31) | 208.02(3.22) | 210.23(2.81) |
| $d_z = 64$ | 7.56(0.4) | 8.56(0.63) | 206.85(5.24) | 207.72(5.66) |

In general, EVAE has the comparable time efficiency to VAE. The slight increase in training time for EVAE results from the sampling process of the Epanechnikov kernel, which requires a few more samples to achieve the Epanechnikov density, as described in Algorithm 2. The difference in training time is negligible when the latent space dimension is large, which facilitates the application of EVAE in high-resolution datasets.

## 6 Discussion and limitation

Approximating the posterior through KDEs, we have derived the optimal kernel in bounding the KL-divergence and built a novel kernel-based VAE called Epanechnikov VAE. The finite support for the Epanechnikov kernel addresses the issue of generating blurry images under Gaussian posterior. Experiments on benchmark datasets demonstrate the power of EVAE compared to vanilla VAE. The result of the presented paper paves the way for various promising research directions for future works.

## 6.1  Connections between EVAE and $\beta$-VAE

When the support parameter B is constant, the target function (10) is

$$\tilde{\mathcal{L}}(\theta, \phi, \mathbf{B}; \mathbf{x}^{(i)}) \approx -\frac{1}{L}\sum_{l=1}^{L}(\log p_\theta(\mathbf{x}^{(i)})|\mathbf{z}^{(i,l)}) + \frac{3B}{5M}\sum_{k=1}^{p}\frac{1}{r_k^{(i)}},$$

where the ratio $\frac{B}{M}$ can also be viewed as a weight parameter in penalizing the kernel term. This formula is similar to the ELBO given in $\beta$-VAE. It would be interesting to compare the disentanglement and implicit regularization effect (Kumar & Poole, 2020) of EVAE and $\beta$-VAE. In fact, in EVAE the weight parameter $\frac{B}{M}$ is derived from the global deviation result for KDEs, i.e. Theorem 3.1, where the constant B should be interpreted as the length of support interval for the prior.

## 6.2  Geometric interpretation of EVAE

Identity (12) provides another interpretation of EVAE. By approximating the posterior with kernel densities, the posterior can be decomposed with the sum of prior and distribution of new information. The Epanechnikov kernel can be viewed as the optimal direction of updating posterior from the prior. This decomposition bridges EVAE with Information Geometry (Nielsen, 2020) which analyzes the relationship between probability distributions by differential geometry.

On the other hand, the limitations of EVAE involve following few points:

## 6.3  A more precise approximation of KL-divergence

In Section 2.2, we bounded the KL-term by a simple inequality $\log t \leq 1 - t$, which limits the more general cases for posteriors and priors. It's possible to derive sharper bounds with higher order approximation where the Epanechnikov kernel may not be the optimal one.

## 6.4  Optimal kernel under different criterions

In this paper, we mainly focused on the $L_2$ deviation of KDEs, which is measured by the functional $I(K)$ proposed in Theorem 3.1. However, in the general theory of KDEs, different criterions lead to different optimal kernels. For example, we didn't put too much attention on the convolution functional $J(K) = \int \left[\int K(t+y)K(t)dt\right]^2 dy$, which is related to the asymptotic variance of statistic $T_m$ defined in section 3. If we want to minimize the asymptotic variance of $T_m$, the optimal kernel is just the uniform kernel, as derived by Ghosh & Huang (1991).

## 6.5  The factorization of approximate posterior

Our derivation of the new target function (10) is based on the independence among hidden dimensions, which is a fairly strong condition in practice. Exploring the theory of KDEs with dependent latent dimensions would be another compelling direction.

## Broader Impact Statement

This paper aims to propose a new perspective in modeling the posterior in generative models by kernel density estimations. The theoretical results should not have negative societal impacts. One possible negative impact resulting from EVAE might be the misuse of generative models in producing fake images which may lead to security issues in some face recognition based systems. Few mitigation strategies: (1) gate the release of models for commercial use; (2) add a mechanism for monitoring fake images generated by models such as the discriminator in GAN models. We can also restrict the private datasets used in training the generative model. All benchmark datasets used in this paper are public and well known to the machine learning community.

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

# A Proofs

## A.1 Proof of Lemma 3.2

For simplicity, we first consider the following constraints

$$K \geq 0, \quad \int K(t)dt = 1, \quad \int tK(t)dt = 0, \int t^2 K(t)dt = \frac{1}{5}.$$

By the method of undermined multipliers, it's equivalent to minimize the following target functional without constraints:

$$\int K^2(t) + aK(t) + ct^2 K(t)dt,$$

with simplified constraints above and $a, c$ are undermined real coefficients. We ignore the term $tK(t)$ as it does not contribute to the unconstrained target function now.

For fixed $t$, denote $y(K) = K^2 + aK + ct^2 K, (K \geq 0)$. Note that the quadratic function $y(K)$ achieves minimum when $K = -\frac{c}{2}t^2 - \frac{a}{2}$.

It follows that $y(K)$ is minimized subject to $K \geq 0$ by

$$K(t) = \begin{cases} -\frac{c}{2}t^2 - \frac{a}{2} & -\frac{c}{2}t^2 - \frac{a}{2} \geq 0 \,. \\ 0 & -\frac{c}{2}t^2 - \frac{a}{2} < 0 \,. \end{cases}$$

We can rewrite it as

$$K(t) = \begin{cases} A(B^2 - t^2) & |t| \leq B \,. \\ 0 & \text{otherwise} \,. \end{cases}$$

for some number $A, B$. By simplified assumptions $\int tK(t)dt = 0, \int t^2 K(t)dt = \frac{1}{5}$, we can find that $A = \frac{3}{4}, B = 1$.

Under general moment conditions in Lemma 3.2, optimal $K^*$ can be written as

$$K^*(t) = \begin{cases} \frac{3}{4r}\left(1 - \left(\frac{t-\mu}{r}\right)^2\right) & |t - \mu| \leq r \,. \\ 0 & \text{otherwise} \,. \end{cases}$$

by location-scale formula. In literature Epanechnikov (1969), this kernel is called Epanechnikov kernel. We put $1/5$ in front of the constraint of second moment in order to make the resulting support interval cleaner, which won't change the optimal kernel. The corresponding optimal value of $I(K)$ is $I(K^*) = \frac{3}{5r}$.

**Plots of Standard Epanechnikov kernel and Guassian kernel**

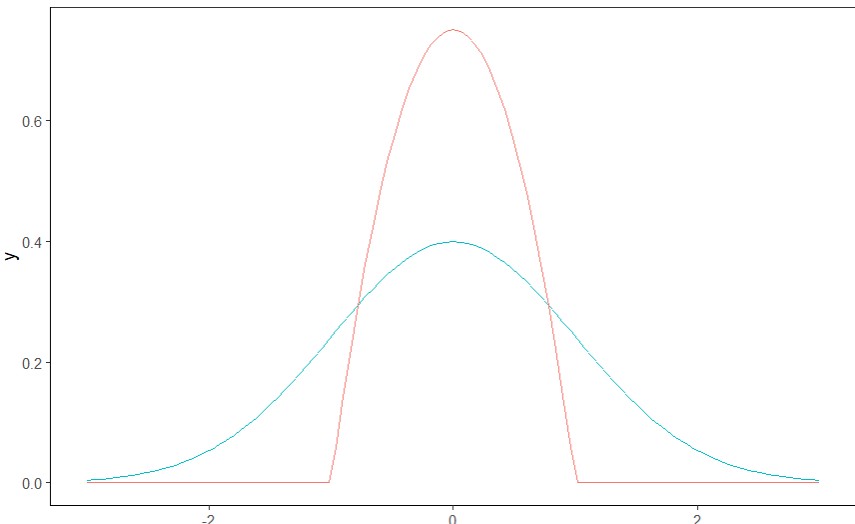

Figure S4: Red curve: Standard Epanechnikov kernel. Green curve: Standard Gaussian kernel.

## A.2 Theoretical support for Algorithm 2

Given $U_1, U_2, U_3 \overset{i.i.d.}{\sim}$ Unif$[-1, 1]$, we only need to show the density of median$(U_1, U_2, U_3)$ is standard Epanechnikov kernel.

Denote $Y = \text{Median}(U_1, U_2, U_3)$, we have

$$
\begin{aligned}
P(Y \leq t) &= P(\text{Median}(U_1, U_2, U_3) \leq t) \\
&= P(\text{Exactly two of } U_1, U_2, U_3 \text{ are less than } t) + P(\text{All of } U_1, U_2, U_3 \text{ are less than } t) \\
&= \binom{3}{2} \left( \frac{1+t}{2} \right)^2 \left( \frac{1-t}{2} \right) + \binom{3}{3} \left( \frac{1+t}{2} \right)^3 \\
&= \frac{1}{2} + \frac{3}{4}t - \frac{t^3}{4} \, .
\end{aligned}
$$

Then the density $f_Y(t)$ of $Y$ is

$$
f_Y(t) = \frac{3}{4} - \frac{3}{4}t^2 = \frac{3}{4}(1 - t^2) \, ,
$$

which is essentially the standard Epanechnikov kernel.

## B Model architecture

### B.1 MNIST and Fashion-MNIST

We used fully convolutional architectures with $4 \times 4$ convolutional filters for both encoder and decoder in EVAE and VAE, as described following. All convolutions in the encoder and decoder employed SAME padding.

We resized images in MNIST and Fashion-MNIST from $28 \times 28$ to $32 \times 32$ at beginning. In the last conv layer, the sigmoid activation function was used to restrict the range of output as we assumed Bernoulli type model of $p_\theta(\mathbf{x}|\mathbf{z})$ and the binary cross entropy loss employed used (reduction to sum). Dimensions $d_z$ for latent space : $\{8, 16, 32, 64\}$

**Encoder $q_\phi$:**

$$x \in \mathbb{R}^{32 \times 32} \to 32 \text{ Conv, Stride } 2 \to \text{BatchNorm} \to \text{ReLU}$$
$$\to 64 \text{ Conv, Stride } 2 \to \text{BatchNorm} \to \text{ReLU}$$
$$\to 128 \text{ Conv, Stride } 2 \to \text{BatchNorm} \to \text{ReLU}$$
$$\to 256 \text{ Conv, Stride } 2 \to \text{BatchNorm} \to \text{ReLU}$$
$$\to \text{Fully connected } (1 * 1 * 256 \times d_z) \text{ for each parameters}$$

**Decoder** $p_\theta$**:**

$$z \in \mathbb{R}^{d_z \times d_z} \to \text{Fully connected } (d_z \times 1 * 1 * 256)$$
$$\to 128 \text{ ConvTran, Stride } 1 \to \text{BatchNorm} \to \text{ReLU}$$
$$\to 64 \text{ ConvTran, Stride } 2 \to \text{BatchNorm} \to \text{ReLU}$$
$$\to 32 \text{ ConvTran, Stride } 2 \to \text{BatchNorm} \to \text{ReLU}$$
$$\to 1 \text{ ConvTran, Stride } 2 \to \text{Sigmoid}$$

## B.2  CIFAR-10

Again, we used fully convolutional architectures with $4 \times 4$ convolutional filters for both encoder and decoder in EVAE and VAE for CIFAR-10 model. In encoder, we employed a layer of Adaptive Average pool filter. Other settings are the same with MNIST and Fashion-MNIST.

**Encoder** $q_\phi$**:**

$$x \in \mathbb{R}^{32 \times 32} \to 32 \text{ Conv, Stride } 2 \to \text{BatchNorm} \to \text{ReLU}$$
$$\to 64 \text{ Conv, Stride } 2 \to \text{BatchNorm} \to \text{ReLU}$$
$$\to 128 \text{ Conv, Stride } 2 \to \text{BatchNorm} \to \text{ReLU}$$
$$\to 256 \text{ Conv, Stride } 2 \to \text{BatchNorm} \to \text{ReLU}$$
$$\to \text{AdaptiveAvgPool2d}$$
$$\to \text{Fully connected } (1 * 1 * 256 \times d_z) \text{ for each parameters}$$

**Decoder** $p_\theta$**:**

$$z \in \mathbb{R}^{d_z \times d_z} \to \text{Fully connected } (d_z \times 1 * 1 * 256)$$
$$\to 128 \text{ ConvTran, Stride } 1 \to \text{BatchNorm} \to \text{ReLU}$$
$$\to 64 \text{ ConvTran, Stride } 2 \to \text{BatchNorm} \to \text{ReLU}$$
$$\to 32 \text{ ConvTran, Stride } 2 \to \text{BatchNorm} \to \text{ReLU}$$
$$\to 3 \text{ ConvTran, Stride } 2 \to \text{Sigmoid}$$

## B.3  CelebA

For CelebA dataset, we used $5 \times 5$ convolutional filters for both encoder and decoder in EVAE and VAE. Simiar to CIFAR-10, we employed a layer of Adaptive Average pool filter before the fully connected layer in encoder. We first scaled images with Center Crop to $140 \times 140$ and resized them to $64 \times 64$.

**Encoder** $q_\phi$**:**

$$x \in \mathbb{R}^{64 \times 64} \rightarrow 64 \text{ Conv, Stride } 2 \rightarrow \text{BatchNorm} \rightarrow \text{ReLU}$$
$$\rightarrow 128 \text{ Conv, Stride } 2 \rightarrow \text{BatchNorm} \rightarrow \text{ReLU}$$
$$\rightarrow 256 \text{ Conv, Stride } 2 \rightarrow \text{BatchNorm} \rightarrow \text{ReLU}$$
$$\rightarrow 512 \text{ Conv, Stride } 2 \rightarrow \text{BatchNorm} \rightarrow \text{ReLU}$$
$$\rightarrow \text{AdaptiveAvgPool2d}$$
$$\rightarrow \text{Fully connected } (1 * 1 * 512 \times d_z) \text{ for each parameters}$$

**Decoder $p_\theta$:**

$$z \in \mathbb{R}^{d_z \times d_z} \rightarrow \text{Fully connected } (d_z \times 8 * 8 * 512)$$
$$\rightarrow 256 \text{ ConvTran, Stride } 1 \rightarrow \text{BatchNorm} \rightarrow \text{ReLU}$$
$$\rightarrow 128 \text{ ConvTran, Stride } 2 \rightarrow \text{BatchNorm} \rightarrow \text{ReLU}$$
$$\rightarrow 64 \text{ ConvTran, Stride } 2 \rightarrow \text{BatchNorm} \rightarrow \text{ReLU}$$
$$\rightarrow 3 \text{ ConvTran, Stride } 2 \rightarrow \text{Sigmoid}$$

## C   Datasets and Training details

We list details for each benchmark dataset in following table

| Datasets | # Training samples | # Hold-out samples | Original image size |
|---|---|---|---|
| MNIST | 60000 | 10000 | 28*28 |
| Fashion-MNIST | 60000 | 10000 | 28*28 |
| CIFAR-10 | 50000 | 10000 | 32*32 |
| CelebA | 162770 | 19867 | 178*218 |

Note that for MNIST,Fashion-MNIST and CIFAR-10, we used default splittings of training sets and testing sets provided in Pytorch (torchvision.datasets). For CelebA, we used default validation set as hold-out samples.

As to the training details, we used same training parameters for all algorithms and datasets, as described in following table

| | |
|---|---|
| Latent space dimensions $d_z$ | 8,16,32,64 |
| Optimizer | Adam with learning rate 3e-4 |
| Batch size | 100 |
| Epochs | 50 |

**Calculation of Sharpness**

We follow the way in Tolstikhin et al. (2018) in calculating the sharpness of an image. For each generated image, we first transformed it into grayscale and convolved it with the Laplace filter $\begin{pmatrix} 0 & 1 & 0 \\ 1 & -4 & 1 \\ 0 & 1 & 0 \end{pmatrix}$, computed the variance of the resulting activations and took the average of all variances. The resulting number is denoted as sharpness (larger is better). The blurrier image will have less edges. As a result, the variance of activations will be small as most activations will be close to zero. Note that we averaged the sharpness of all reconstructed images from hold-out samples for each dataset.

**Binomial test for two models**

If EVAE and VAE have similar performance in FID score, the probability that EVAE has lower FID score should be 0.5 in each independent experiment. (Same hypothesis for sharpness). However, according to Table

1 and 2, EVAE wins 15 experiments for FID score among all datasets. The p-value of winning 15 experiments under null hypothesis is

$$P(X \geq 15) = \binom{16}{16}(0.5)^{16} + \binom{16}{15}(0.5)^{16} \approx 2.6 \times 10^{-4} < 0.05.$$

P-value is smaller than 0.05 significance level thus EVAE significantly outperforms VAE in FID. Similar calculation can be applied to sharpness, whose p-value of winning 12 experiments is $P(X \geq 15) = \sum_{i=15}^{16} \binom{16}{i} 0.5^{16} \approx 2.6 \times 10^{-4} < 0.05$ and we achieved the same conclusion for the significance of EVAE's superiority in sharpness.

## D  Sampled reconstructed images

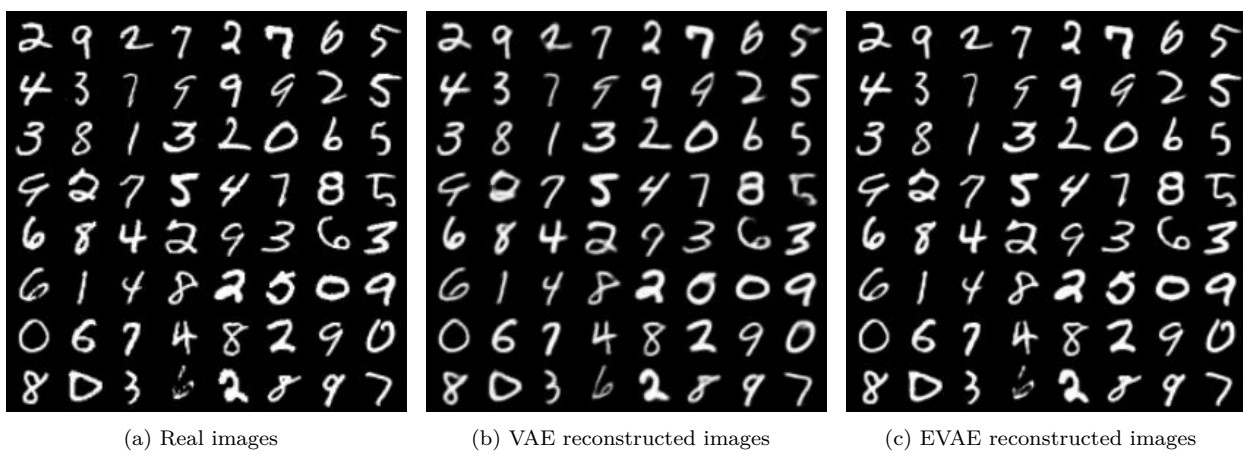

| (a) Real images | (b) VAE reconstructed images | (c) EVAE reconstructed images |

Figure S5: (a) Sampled real images from hold-out samples in MNIST (b) Reconstructed images by VAE. (c) Reconstructed images by EVAE. Dimension $d_z = 64$ for both models. See section B for Model architectures.

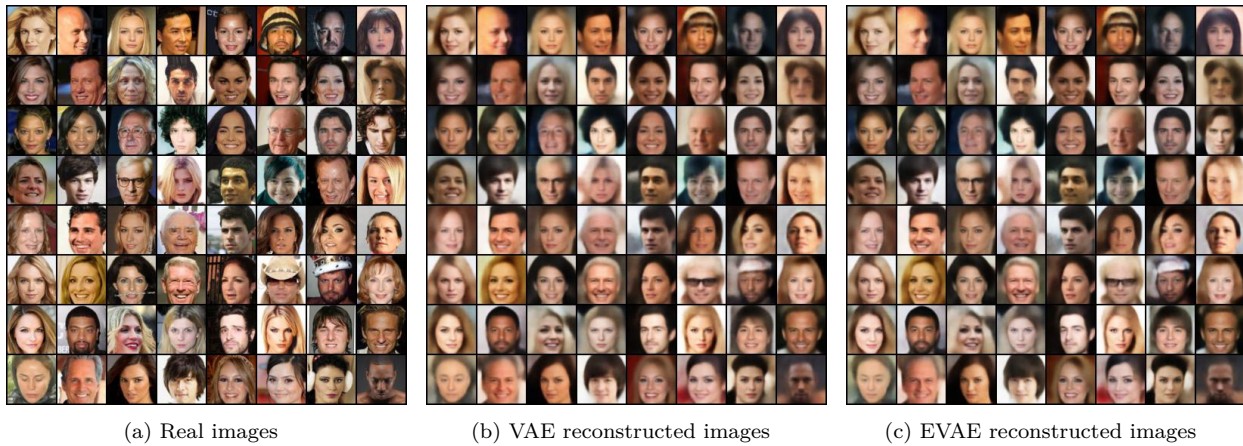

| (a) Real images | (b) VAE reconstructed images | (c) EVAE reconstructed images |

Figure S6: (a) Sampled real images from hold-out samples in CelebA (b) Reconstructed images by VAE. (c) Reconstructed images by EVAE. Dimension $d_z = 64$ for both models.

## E  EVAE with Guassian prior

The experiment details are the saem with section B. The only difference is we replaced uniform prior with Gaussian prior and we still kept the support parameter $B(= 0.1)$ in front of Gaussian prior.

Table 6: VAE and EVAE results on MNIST and Fashion-MNIST datasets

| Datasets | MNIST | | | | Fashion-MNIST | | | |
| --- | --- | --- | --- | --- | --- | --- | --- | --- |
| | VAE | | EVAE | | VAE | | EVAE | |
| $d_z$ | FID | Sharpness | FID | Sharpness | FID | Sharpness | FID | Sharpness |
| 8 | 16.14 | 0.0217 | 15.22 | 0.0246 | 42.22 | 0.0157 | 35.89 | 0.0181 |
| 16 | 12.11 | 0.0248 | 10.25 | 0.0308 | 38.88 | 0.0167 | 26.65 | 0.0218 |
| 32 | 12.43 | 0.0247 | 8.22 | 0.0335 | 39.14 | 0.0157 | 18.52 | 0.0246 |
| 64 | 11.73 | 0.0248 | 5.74 | 0.0364 | 38.96 | 0.0159 | 12.18 | 0.0271 |

Table 7: VAE and EVAE results on CIFAR10 and CelebA datasets

| Datasets | CIFAR-10 | | | | CelebA | | | |
| --- | --- | --- | --- | --- | --- | --- | --- | --- |
| | VAE | | EVAE | | VAE | | EVAE | |
| $d_z$ | FID | Sharpness | FID | Sharpness | FID | Sharpness | FID | Sharpness |
| 8 | 228.92 | 0.0353 | 218.84 | 0.0355 | 194.36 | 0.0149 | 194.62 | 0.0145 |
| 16 | 183.03 | 0.0360 | 163.84 | 0.0355 | 144.88 | 0.0155 | 147.78 | 0.0157 |
| 32 | 146.80 | 0.0359 | 122.17 | 0.0364 | 106.02 | 0.0160 | 102.73 | 0.0163 |
| 64 | 141.22 | 0.0353 | 79.06 | 0.0356 | 80.60 | 0.0161 | 65.79 | 0.0164 |

# F   Validation curves

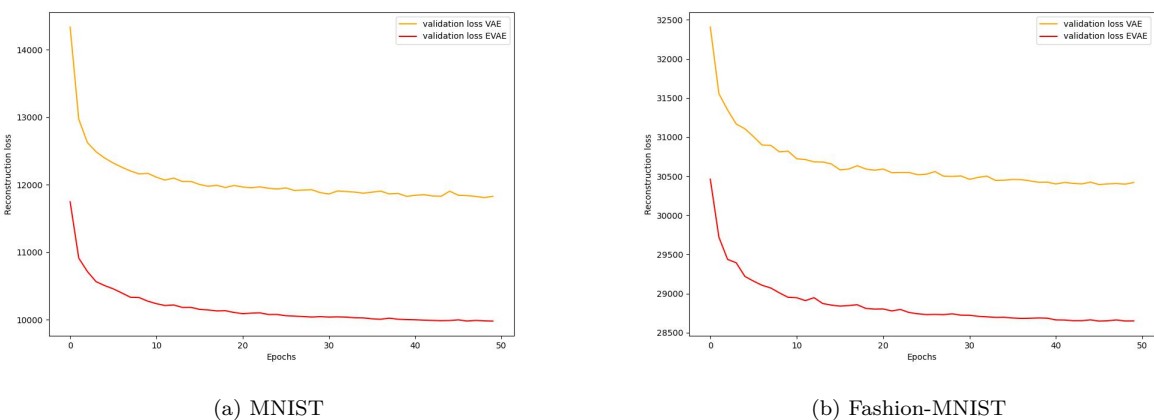

(a) MNIST

(b) Fashion-MNIST

Figure S7: Reconstruction validation loss curves as function of Epochs. ($d_z = 64$). Red curve is for EVAE and yellow one represents VAE. Binary cross entropy loss is reduced to sum for each batch (a) MNIST (b) Fashion-MNIST.

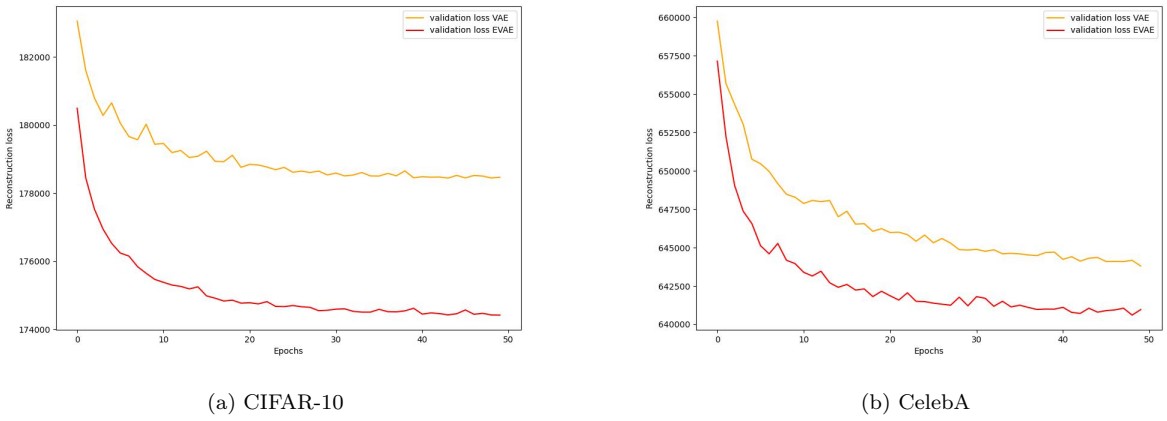

(a) CIFAR-10

(b) CelebA

Figure S8: Reconstruction validation loss curves as function of Epochs. ($d_z = 64$). Red curve is for EVAE and yellow one represents VAE. Binary cross entropy loss is reduced to sum for each batch (a) CIFAR-10 (b) CelebA.

