# OpenReview forum: "Epanechnikov Variational Autoencoder"
_TMLR — Rejected by TMLR_

### Review · Reviewer_Lx4G · 2025-03-01

**Summary Of Contributions:**

This paper bridges Variational Autoencoders (VAE) and kernel density estimations (KDE) by approximating the posterior using KDE, leading to a new lower bound of empirical log likelihood. The approach overcomes the limitations of Gaussian latent space in vanilla VAE and offers a novel perspective on estimating the KL-divergence term in the ELBO. The paper introduces the Epanechnikov kernel-based VAE (EVAE), which improves sample quality by reducing noise and blurriness compared to the Gaussian kernel. Experiments on benchmark datasets demonstrate the superior performance of EVAE in reconstructing images, as measured by FID score and sharpness.

**Audience:**

Yes

**Claims And Evidence:**

Yes

**Requested Changes:**

1. Check the notation throughout the paper and distinguish between scalars, vectors, and matrices.
   - Scalars: Use regular text or math mode without bold.
   - Vectors: Use bold lowercase letters (e.g., $\mathbf{z}$).
   - Matrices: Use bold uppercase letters (e.g., $\mathbf{A}$).

   Ensure consistency in notation and make sure they are well-defined, preferably in a notation section at the beginning.

2. Standardize the formatting of all tables in the paper.
   - Use the `booktabs` package for consistent table formatting.
   - Ensure consistent column alignment and layout across all tables.

3. Standardize the capitalization of section titles and fix other capitalization issues in the main body text.

**Strengths And Weaknesses:**

Strengths:

Overall, the paper presents a clear main idea and is relatively smooth to read. The core idea of approximating the posterior using KDEs with Epanechnikov kernels is somewhat innovative, although many conclusions are derived from Rosenblatt (1956). The mathematical derivations are solid, with no apparent flaws or errors.

Weaknesses:

The main drawback of the paper is the roughness in writing. While the overall structure and logic are sound, many small details suggest a lack of thorough preparation. For example, the notation for $\mathbf{z}$ is not consistently written in bold, and the formatting of tables (e.g., Table 3) is inconsistent. Additionally, the capitalization of section titles is not standardized. These are minor issues but should be addressed for improved clarity and professionalism.

---

> ### Author Response · Authors · 2025-03-12
> **Thank you for the detailed feedback**
>
> Dear reviewer,
>
> Thank you for the detailed suggestions. We are glad you agree that approximating the posterior using KDEs with Epanechnikov kernels is somewhat novel.
>
> We have updated the manuscript accordingly. More specifically,
> * The notation of $\bf{z}$ is now in bold for consistency.
> * All tables (including Table3) are formatted by booktabs with consistent layout.
> * The capitalization of section titles are standardized according to the TMLR template. We also make the in-text titles in section 1 and 6 such as limitations of EVAE as subsection titles with appropriate capitalization.
>
> The updated manuscript will be uploaded as soon as possible after we receive all reviews.

---

> > ### Author Response · Authors · 2025-03-16
> >
> > Dear reviewer,
> >
> > The updated pdf has been uploaded according to your suggestions. All changes are highlighted in red color. We hope our modifications address your concerns in the clarity and professionalism of our work.

---

> > > ### Comment · Reviewer_Lx4G · 2025-03-26
> > >
> > > Thank you to the authors for their response. The authors have addressed my concern, and I am satisfied with their reply.

---

### Review · Reviewer_L8JT · 2025-03-12

**Summary Of Contributions:**

The vanilla VAE uses 1) Gaussian encoder (variational distribution) + 2) Gaussian decoder + 3) Gaussian prior. There are obvious limitations on the expressiveness of the vanilla VAE. This paper proposes the EVAE which uses a non-Gaussian encoder (variational distribution) based on KDE and a uniform prior. The EVAE variational distribution is derived to minimize a bound on the KL term in the ELBO assuming a variational family of suitably regular kernel density estimators.

**Audience:**

Yes

**Broader Impact Concerns:**

None.

**Claims And Evidence:**

No

**Requested Changes:**

Major comments:

1. I cannot vote for acceptance unless my concern under Weakness is addressed. One way to address it might be to drop all mention or insinuation that the EVAE is optimal in some sense. You can simply set your variational family to be based on Epanechnikov KDEs. Then everything else still goes through.
2. I also would like to see the quality of samples generated from a trained EVAE, i.e., sample the latent $z$ from your uniform prior, and then run it through the learned decoder.

Minor comments (mostly typos and easily fixable mistakes):

* Many math expressions are missing a space after $\\log$
* $b(m)$ in the KDE first used at the bottom of Pg. 3 is never defined. Earlier you use $b\_n$ to introduce the general KDE.
* You might consider using $m$ instead of $n$ in your introduction to KDEs at the beginning of 2.2.1 to match up with the notation in $q\_{m,\\phi}$
* The prior on the latent variable is first introduced as parametric $p\_\\theta(z)$ at the beginning of Section 2.1. But the $\\theta$ is dropped immediately.
* Eqn (2) and Eqn (1) should only differ by a sign, but Eqn (2) uses a different expectation measure
* There needs to be space between $I(K)$ and $J(K)$ in Eqn 7
* $B$ is introduced and defined under Eqn (8) as the length of support interval of prior $p(z)$ under the $i$-th data point. I need a mathematical definition. From the looks of it, $B= \\int 1 \\,dz$ and I fail to see how this depends on $i$. \[Revision: Ok after reading futher, I gather you meant $B$ depends on the latent variable dimension.\]
* Above Eqn 8, “the inequality equation 5 becomes” is not using equation reference environment consistently with the rest of the paper.
* Above Lemma 3.2 it is written “we can now find an optimal kernel function $I(K_{\phi, x^{(i)} })$ which minimizes $I(K_{\phi, x^{ (i) }} )$
* It is not clear the role of the encoding networks with parameters $\\phi$ and how they produce $\\mu^{(i)}$ and $r^{(i)}$. I think the issue stems from that fact that you never defined $K\_\\phi^\*$ in Eqn 9\. \[I can guess what you mean but I’d rather not.\]
* The prior used in EVAE is set to the uniform distribution which is discussed right above Algorithm 1\. I think this fact needs a few more reminders, in particular in the experiment section.
* Please check that the indexing is correct throughout. For instance, in Eqn (5), the dependence on the $i$-th data point is explicit. But Eqn (6) should not have this dependence and yet the very last expression depends on $i$ again.

**Strengths And Weaknesses:**

Strengths:
The EVAE paper contributes a new pair of variational distribution $q_\phi(z|x)$ and prior $p(z)$ where the KL divergence between them $KL(q_\phi(z|x) || p(z) )$ can be derived in closed form. This is supposed to overcome the limitation of complex variational and prior pairs not having closed form KL divergence.

Weakness:
The paper introduces a further bound on the ELBO arriving at
$$
\log p_\theta(x) \ge E_{q_\phi(z|x)} \log p_\theta(z|x) - E_0 \int (q_{m,\phi}(z) - p(z))^2 /p(z) \,dz
$$
where the variational distribution is based on a KDE $q_{m,\phi}$. The paper then goes to some lengths to derive the optimal kernel in the KDE such that the second term above is minimized.

But what justifies ignoring the first term? With the same rationale, why don’t we just set the variational distribution to the prior exactly, then we can minimize the KL term in the ELBO.

The overall optimization must balance reconstruction error and closeness to the prior, and these are conflicting objectives. It simply doesn’t make sense to me that it’s enough to find the variational distribution that minimizes the second term while ignoring the first term.

---

> ### Author Response · Authors · 2025-03-12
> **Response to weakness (major comment 1)**
>
> Dear reviewer,
>
> Thank you for your detailed review and we are glad that you recognize the flexibility of EVAE. We will first address your major concern as follows.
>
> To be clear, $q_{m,\phi}(z)$ is a kernel density estimator of posterior $q_{\phi}(\mathbf{z}|\mathbf{x}^{(i)})$. More specifically, on page 3 you can see the definition $q_{\phi}(\mathbf{z}|\mathbf{x}^{(i)})=E_{0}(q^{(i)}_{m,\phi}(\mathbf{z}))$ where
>
> $$
> q^{(i)}_{m,\phi}(\mathbf{z})=\frac{1}{mb(m)} \sum _{j=1}^{m} K _{\phi,\mathbf{x}^{(i)}} \left[ \frac{\mathbf{z}-\mathbf{Z} _{j}}{b(m)}\right],\quad \mathbf{Z} _{j}  \overset{i.i.d}{\sim} p(\tilde{\mathbf{z}})
> $$
>
> The lower bound on ELBO you listed is not the whole story . More specifically, by Theorem 3.1 and inequality (8), we have
>
> $$
> KL(q_{\phi}(\mathbf{z}|\mathbf{x}^{(i)})||p(\mathbf{z}))\leq  B\frac{I(K_{\phi,\mathbf{x}^{(i)}})}{n}
> $$
>
> Therefore, our lower bound on ELBO (By the way, there is a typo in your review) becomes
> $$
> \text{log}p_{\theta}(\mathbf{x})\geq  E _{q _{\phi}(\mathbf{z}|\mathbf{x})} [\text{log}p \_{\theta}(\mathbf{x}|\mathbf{z})]-KL(q _{\phi}(\mathbf{z}|\mathbf{x})||p(\mathbf{z}))p(\mathbf{z})) \geq E _{q _{\phi}(\mathbf{z}|\mathbf{x})}  [\text{log}p _{\theta}(\mathbf{x}|\mathbf{z})]- B\frac{I(K _{\phi,\mathbf{x}^{(i)}})}{n}
> $$
> when $n$ large.
>
> Thus we can see the second term only involves the form of kernel $K$ . As we emphasized in introdcution section, we are interested in deriving the optimal **functional kernel from** in controlling the lower bound of ELBO.  Note that the optimal functional form is different from the whole neural network optimization process. The derivation of optimal functional form is purely in mathemetical sense, which introduces the Epanechnikov  kernel as a result. Lemma 3.2 points out that Epanechnikov  kernel enjoys some theoretical optimality.
>
> Once we plug optimal kernel form (Epanechnikov  kernel)in the lower bound of ELBO, which is the new target function in EVAE, the parameters in first term and second term will be optimized together in the neural network optimization process. To this sense, the training of EVAE indeed involves those two terms.
>
> In addition, the first term $E _{q _{\phi}(\mathbf{z}|\mathbf{x})} [\text{log}p \_{\theta}(\mathbf{x}|\mathbf{z})]$  typically refers to the reconstruction error and decoder process $p \_{\theta}(\mathbf{x}|\mathbf{z})$ is usually modeled by gaussian distribution or  Bernoull distribution. In our experiments, we set it to be  gaussian distribution. See [1]. The choice of decoder distribution is independent of our derivation of  optimal functional kernel form in the second term.
>
> Note that the KL divergence of uniform prior and Epanechnikov  kernel has no closed form (i.e. the KL term will a  piecewise function therefore we can't simply set our variational family to be based on Epanechnikov KDEs. Otherwise, the target function of training EVAE can be tricky.
>
>
> [1]Diederik P. Kingma and Max Welling. Auto-encoding variational bayes. CoRR, abs/1312.6114, 2013.

---

> > ### Author Response · Authors · 2025-03-16
> > **Follow up the response to weakness (major comment 1)**
> >
> > Dear reviewer,
> >
> > We have uploaded updated pdf according to your suggestions. All changes are highlighted in red color. More specifically,
> >
> > * We rephrased the idea of finding the optimal kernel in an upper bound of KL-divergence to avoid potential confusion between functional optimization and neural network parameter optimization. In our previous response, we emphasized that we are not obtaining optimal functional form of posterior and prior in KL-divergence. In essence, the quantity we are interested in is the quadratic functional I(K), which appears in the upper bound of KL-divergence we derived. By finding the kernel minimizing the quantity I(K), the corresponding upper bound should be the tightest one among other validate kernel candidates, given parameters and data point.  Thus the Epanechnikov kernel is theoretical optimal in the sense of minimizing the quadratic functional I(K), which is supported by Lemma 3.2.  This result has an interesting geometric interpretation as we pointed out in the discussion part:  "The Epanechnikov kernel can be viewed as the optimal direction of updating posterior from the prior."
> >
> > We understand your concern and in the updated pdf, we made the words of functional optimality of Epanechnikov kernel as specific as possible to avoid possible confusion for readers. For example, we said "Epanechnikov kernel gives the tightest upper bound of KL-diverngence given parameters and data point" instead of "Epanechnikov kernel is optimal in controlling KL-divergence".
> >
> > Additionally, the Epanechnikov kernel has closed form in the quadratic functional I(K) while it typically has no closed form in KL-divergence with respect to other priors, which is another motivation of introducing KDE in VAE. By utilizing the inequality (8), we can have closed form of target function in EVAE. We hope our explanations clarify the misunderstanding.
> >
> > * All minor comments have been addressed. As to the reason why B could rely on the i-th data point, our explanation is: (1)In the most general case, different data points could come from different prior distributions, leading to different length of support interval. (2)  Your understanding that B could depend on the latent variable dimension is also correct as in multi-dimension case, the latent distribution may have different marginal distribution.  For simplicity, we assume B is constant for all datapoints and latent variable dimention and have added a corresponding sentence in updated manuscript.
> >
> > * We also want to reiterate that the quality of samples generated from a trained EVAE is already displayed in section 5.5 (Unconditional samples) of the initial manuscript. We hope this clarification would address your concern.

---

> > > ### Comment · Reviewer_L8JT · 2025-04-02
> > > **Response to rebuttal**
> > >
> > > Thank you for the detailed response. I'm glad we found some common ground on the phrasing around "optimality".
> > >
> > > I still hold some reservations on the overall motivation. You want a variational family that is a) rich and b) still has closed form KL(variational distribution||prior). But in essence, you replace the KL term with an upper bound and it is the upper bound that has a closed form. Then I would argue it's simply more straightforward to do empirical estimation of (b) rather than a closed-form of an upper bound of (b). I believe rQoF shares a similar concern.
> > >
> > > I do like the other feature that the support need not match or be nested in a certain way. But this feature is contingent on the choice of B which seems understudied. If this is a main feature of the proposed methodology, I would've liked to see more literature review on how existing VAE methods deal with the support issue, which I believe is a real issue.

---

> > > > ### Author Response · Authors · 2025-04-02
> > > > **Thank you for the feedback.**
> > > >
> > > > Dear reviewer,
> > > >
> > > > Thank you for the additional feedback. We will address your concerns as follows:
> > > >
> > > > **1. Response to the sentence "You want a variational family that is a) rich and b) still has closed form KL(variational distribution||prior)."**
> > > >
> > > > In the introduction of the updated pdf, we stated that "Most variants of VAE along these two directions have the closed-form KL divergence. This is essential in implementation but somehow limits the potential applications of more flexible pairs of posterior and prior which has no closed-form of KL divergence but could better capture the latent data distribution and alleviate
> > > > model collapse...".  And  our motivation is to alleviate the obstacle of closed form KL((variational distribution||prior)) by replacing the KL term with an upper bound which has a closed form for many distributions. In other words, we want to relax the requirement of closed form KL term in VAE.
> > > >
> > > > Therefore, we actually want a variational family that is a) rich and b) **has a closed upper bound of the KL(variational distribution||prior)**
> > > >
> > > > **2. Response to "...do empirical estimation of (b) rather than a closed-form of an upper bound of (b)..."**
> > > >
> > > > As we responded to the reviewer rQoF, for complicated posterior, it's indeed common to employ Monte-Carlo simulation (MC) to approximate KL. But MC can be very time consuming for complex distributions. However, in EVAE, the sampling step is time efficient as described by Algorithm 2. And section 5.7 demonstrated that EVAE has a decent time efficiency compared to VAE, which is one of the advantages of EVAE. We emphasized this point in the updated introduction (See bullet 4 in the list of main contributions)
> > > >
> > > > Even if we can design a complicated pair of posterior and prior which indeed has closed form of KL, the sampling process can be tricky and time consuming as well. For example, [1] proposed using a von Mises-Fisher (vMF) distribution instead of Gaussian to better capture latent hyperspherical structure. With uniform prior, their proposed Hyperspherical Variational Auto-Encoders(HVAE) have closed form of KL while sampling from von Mises-Fisher (vMF) distribution requires acceptance-rejection method, which can be time consuming for high dimensional latent space.
> > > >
> > > > [1] Tim R. Davidson, Luca Falorsi, Nicola De Cao, Thomas Kipf, and Jakub M. Tomczak. Hyperspherical
> > > > variational auto-encoders. ArXiv, abs/1804.00891, 2018
> > > >
> > > > **3. Literature review on how existing VAE methods deal with the support issue**
> > > >
> > > > We are glad you appreciate the support of EVAE need not match or be nested in a certain way. As we stated in **section 6.1 Connections between EVAE and β-VAE",  when the support parameter B is constant, our target function becomes
> > > >
> > > > $$
> > > >   \tilde{\mathcal{L}}(\mathbf{\theta},\mathbf{\phi},\mathbf{B};\mathbf{x}^{(i)})\approx -\frac{1}{L}\sum_{l=1}^{L}(\text{log}p_{\theta}(\mathbf{x}^{(i)})|\mathbf{z}^{(i,l)})+\frac{3B}{5M}\sum_{k=1}^{p}\frac{1}{r_{k}^{(i)}}
> > > > $$
> > > >
> > > > where the ratio B/M (M is the batch size) can also be viewed as a weight parameter in penalizing the kernel term.  This formula is similar to the ELBO given in β-VAE [2].  It would be interesting to compare the **disentanglement** and **implicit
> > > > regularization effect** [3] of EVAE and β-VAE, which would be our future researches. In fact, in EVAE the weight parameter B/M is derived from the global deviation result for KDEs, i.e. Theorem 3.1, where the constant B should be interpreted as the length of support interval for the prior.
> > > >
> > > >  β-VAE [2] demonstrates that increasing β encourages disentanglement of latent factors, where $\beta$ is a coefficient in front of KL-term, which plays a similar trade-off role of support B in EVAE.  [4] Introduces a progressive  increases the information capacity of the latent code during training to better trade-off between reconstruction and disentanglement. [5] interprates the $\beta$ in the view of multual information and demonstrates that increasing $\beta$ leads to lower mutual information, which can degrade reconstruction. Their conclusion is consistent with our observation of support B in section 5.2, which implies the potential connection between $B$ and $\beta$.
> > > >
> > > >
> > > > [2] Irina Higgins, Loïc Matthey, Arka Pal, Christopher P. Burgess, Xavier Glorot, Matthew M. Botvinick,
> > > > Shakir Mohamed, and Alexander Lerchner. beta-vae: Learning basic visual concepts with a constrained
> > > > variational framework. In International Conference on Learning Representations, 2016.
> > > >
> > > > [3] Abhishek Kumar and Ben Poole. On implicit regularization in β-VAEs. In Hal Daumé III and Aarti Singh
> > > > (eds.), Proceedings of the 37th International Conference on Machine Learning,
> > > >
> > > > [4] Burgess, C. P., Higgins, I., Pal, A., Matthey, L., Watters, N., Desjardins, G., & Lerchner, A. (2018). "Understanding Disentangling in β-VAE", arXiv:1804.03599
> > > >
> > > > [5] Zhao, S., Song, J., & Ermon, S. (2017). "InfoVAE: Information Maximizing Variational Autoencoders", arXiv:1706.02262

---

> ### Author Response · Authors · 2025-03-12
> **Response to other comments**
>
> Dear reviewer,
>
> The quality of samples generated from a trained EVAE is displayed in section 5.5 (Unconditional samples).  We observe that the larger the value of B is, the more diversed novel samples will be generated from EVAE. This makes sense as smaller value of B will make
> the model focus more on the perturbation part (KDE based) of identity (12), leading to the prior collapse.To this point, we can view the parameter B as a way to trade off the reconstruction and sample diversity.
>
> As you suggested in the section of minor comments, we will modify the typos and mistakes accordingly. The updated pdf will be uploaded as soon as possible after we receive all reviews. Thank you again for the detailed suggestions.

---

> > ### Comment · Reviewer_L8JT · 2025-04-02
> > **Apologies for missing the generated samples in section 5.5**
> >
> > Thanks for correcting my oversight.

---

### Review · Reviewer_rQoF · 2025-03-16

**Summary Of Contributions:**

The paper proposes to use a more flexible posterior within a VAE given as an expectation of a Kernel density estimator. A practical implementation based on an upper bound of the KL-divergence and a particular Kernel that works well with that upper bound are presented. The method is evaluated on some image datasets and found to be superior to standard VAE variants.

**Audience:**

No

**Broader Impact Concerns:**

There are no concerns about ethical implications.

**Claims And Evidence:**

No

**Requested Changes:**

1. I encourage the authors to fix all typos, and run a spellchecker. Independent of content, the paper requires quite a bit of polishing to be suitable for TMLR (critical for securing my recommendation)
2. The writing of the introduction needs to be improved (see weaknesses). Most importantly, it needs immediately clear what problem the method addresses.  There are other ways of addressing blurry samples in VAEs, why are they not enough? (critical for securing my recommendation)
3. Situate the work within modern machine learning and why it is important (e.g., in the introduction). Only few references newer than 2018 are given.  (would strengthen the work)
4. Better motivate the choice of posterior approximation, why not simply use a truncated Gaussian location scale family if sharper samples is what we care about?  (would strengthen the work)

**Strengths And Weaknesses:**

Strengths:
- The empirical results seem promising, and the proposed method works better than VAE in several metrics. Samples on CIFAR-10 seems sharper, which supports some of the claims.

Weaknesss:
- The natural question "In which sense of optimality there exists an optimal functional form of posteriors given certain technical conditions?" did not seem very natural to me, and I could not understand it or what kind of problems it tries to address.
- Truncated Gaussians are in location-scale family, why not simply use them rather than taking the long route through the KDE? Evaluation of 1D integrals in KL divergence should not be a problem (e.g. using quadrature)
- The writing of the paper can be improved for a better flow. Specifically:
1) the introduction dives directly into details about VAEs, but broader challenges are not addressed and the paper is not situated within the recent literature (almost no citations newer that 2018 in the intro)
2) some details are not introduced at the right time, for instance, when reading the introduction the reader may not be interested in the details of GPU and computer the experiments are run on. this could go into the experiments section.
3) it is written that to maximize the empirical log likelihood, intractable integrals are required, and then in the next paragraph it is mentioned that it is not required
- The KL divergence cannot be computed exactly, and it remains unclear what is lost by the bound
- I was unable to understand the E_0 notation.

Typos:
- Gaussian -> Gaussian on pg.2
- there seems to be a bit of space missing between log and p_\theta, in general a space after \log is needed.
- evidende -> evidence
- there is a comma after Eq2 sitting in a new line
- punctuation needs to be added to equations
- on top of page 4, the word kernel is suddenly capitalized
- Eq. 6 exceeds the line width
- Section 4 title, Epanechniko -> Epanechnikov?
- Check the capitalization of names, e.g., 5.4 gaussian prior -> Gaussian prior
- the capitilization of words in the papers in the reference list is often wrong, citation styles are mixed and require cleaning up / simplification

---

> ### Author Response · Authors · 2025-03-17
> **Response to requested changes**
>
> Dear reviewer,
>
> Thank you for your constructive feedback and we are glad you agree that our empirical results are promising. We will address your requested changes (along with weakness) as follows:
>
> **1. Fixing all typos and polishing the paper**
>
> **Response:**
>
> We have updated pdf which addresses all typos as you listed in the review. All changes are highlighted in red color. Additionally, the details of GPU and computer the experiments are run on have been put in experiment section.  We hope the updated pdf could address your concern
>
> **2. Improving the writing of the introduction**
>
> * We have rewrote the introduction part by replacing the initial question "In which sense of optimality there exists an optimal functional form of posteriors given certain technical conditions?" by the problems we want to address. More specifically, essentially we want to utilize the flexibility of KDE to enrich the current VAE variants as most methods still exploit the distributions with closed form of KL-divergence. This would limit the applications of more general distributions. Our logic flow is deriving an upper bound of KL-divergence which is relatively easier to evaluate and has closed form for many distributions to avoid this obstacle.
>
> * Additionally,  the quantity we are interested in is the quadratic functional I(K) instead of KL-divergence, which appears in the upper bound of KL-divergence we derived. By finding the kernel minimizing the quantity I(K), the corresponding upper bound should be the tightest one among other validate kernel candidates, given parameters and data point. Thus the Epanechnikov kernel is theoretical optimal in the sense of minimizing the quadratic functional I(K), which is supported by Lemma 3.2.  Finding the kernel K minimizing the quantity I(K) is essentially a functional (theoretical) optimization problem and this is different from the neural network optimization. We hope this explanation could clarify your initial confusion about the question we are trying to answer in the original manuscript.
>
> * Another possible misunderstanding is, we **didn't aim** to address blurry samples in VAE. The sharper samples from EVAE is just a byproduct of introducing Epanechnikov kernel in VAE not our original goal in this paper. We used the sharpness to evaluate the quality of reconstructed samples from EVAE and support our claim that distributions with compact support could generate sharper samples. That is to say, addressing blurry samples in VAE is out of the scope of our current paper.
>
> **3. Situate the work within modern machine learning**
>
> **Response:**
>
> We have added few more current references in section 1.1 and 1.2. More specifically, [1] proposed the Aggregate Variational Autoencoder (AVAE), which estimates the posterior by Gaussian kernel density estimations (KDEs) to prevent issues like
> posterior collapse and improve the quality of the learned latent space. For data with
> heavy-tailed characteristics, [2] replaced Gaussian prior with heavy-tailed distributions such as the Student’s t-distribution, which allows the model to capture data with higher kurtosis, leading to more robust representation. [3] utilized optimal transport theory design priors that enforce a coupling between the prior and data distribution.
>
> However methods are still along the direction of closed-form KL-divergence, which restricts the application of more general distributions.  To improve the flexibility of the choice of posterior and prior in VAE, in this paper, we estimate the posterior by the expectation of KDEs and derive a corresponding upper bound of KL-divergence, which has closed-form for many distributions. Such flexibility also makes the
> derivation of the optimal functional form of kernel possible. Note that [1] directly employed a Gaussian kernel density estimator to estimate the posterior, which requires large number of samples to achieve decent performance.
>
> [1] Surojit Saha, Sarang C. Joshi, and Ross T. Whitaker. Matching aggregate posteriors in the variational
> autoencoder. In ICPR (6), pp. 428–444, 2024.
>
> [2] Juno Kim, Jaehyuk Kwon, Mincheol Cho, Hyunjong Lee, and Joong-Ho Won. $t^3$-variational autoencoder:
> Learning heavy-tailed data with student’s t and power divergence. In The Twelfth International Conference
> on Learning Representations, 2024.
>
> [3] Xiaoran Hao and Patrick Shafto. Coupled variational autoencoder. In Andreas Krause, Emma Brunskill,
> Kyunghyun Cho, Barbara Engelhardt, Sivan Sabato, and Jonathan Scarlett (eds.), Proceedings of the 40th
> International Conference on Machine Learning, volume 202 of Proceedings of Machine Learning Research,
> pp. 12546–12555. PMLR, 23–29 Jul 2023
>
> We hope our rewrote introduction addresses your concern.

---

> > ### Author Response · Authors · 2025-03-17
> > **Response to requested changes (continued)**
> >
> > **4. "Better motivate the choice of posterior approximation, why not simply use a truncated Gaussian location scale family if sharper samples is what we care about?"**
> >
> > **Response:**
> >
> > * First of all, getting sharper samples is not the goal in our work. We aims to introduce a new type of KDE based posterior in VAE rather than address the blurry samples. Sharper samples from KDE is just a byproduct.
> > * Second, unfortunately, truncated Gaussian location scale family has no closed-form of KL-divergence  in general due to the support intervals. Nielsen[1] only reported closed-form formulas for calculating the KL divergence  between two truncated normal distributions where the support of the first distribution is **nested** into the support of the second distribution. However, in practice, this assumption is not guaranteed, which makes the target function tricky. The piecewise target function can be unstable in training a neural network. However, the quadratic functional I(K) introduced in inequality (8) has closed form for many distributions, including truncated Gaussian. We don't need to worry about the support interval of distributions as this information is summarized in parameter B during our derivation.
> >
> > In addition, theoretically, by lemma 3.2 we have showned that Epanechnikov kernel is the optimal kernel in minimizing the functional I(K), which further motivates the choice of posterior approximation.
> >
> > [1]Nielsen F. Statistical Divergences between Densities of Truncated Exponential Families with Nested Supports: Duo Bregman and Duo Jensen Divergences. Entropy (Basel). 2022;24(3):421. Published 2022 Mar 17.
> >
> > **Other weakness 1: "it is written that to maximize the empirical log likelihood, intractable integrals are required, and then in the next paragraph it is mentioned that it is not required"**
> >
> > **Response:**
> >
> > That is because the  intractable integrals in log evidence could be avoided by maximizing the corresponding log evidence lower bound (ELBO), which has been explained in equation (1). This is also the fundamental idea behind the VAE.
> >
> >
> > **Other weakness 2: "The KL divergence cannot be computed exactly, and it remains unclear what is lost by the bound"**
> >
> > **Response:**
> >
> > That is a good question and we actually listed this point in the discussion part. Essentially, we can obtain a more accurate bound of KL divergence by its higher order Taylor expansion. This is one of our future works. Inequality (4) can actually be derived from the first order Taylor expansion of KL-divergence.
> >
> > **Other weakness 3: "I was unable to understand the E_0 notation."**
> >
> > **Response:**
> >
> > Right behind equation (4), we mentioned that $E_{0}$ is the expecation taken w.r.t the samples from  prior distribution $p(\tilde{\mathbf{z}})$. More specifically, given kernel $K$ you can understand $E_{0}$ as follows:
> >
> > $E_{0}(q_{m,\phi}(\mathbf{z}))=E_{0}\left[\frac{1}{mb(m)}\sum \_{j=1}^{m}K _{\phi,\mathbf{x}}\left( \frac{\mathbf{z}-\mathbf{Z} _{j}}{b(m)}\right) \right]=E _{0}\left[\frac{1}{b(m)}K _{\phi,\mathbf{x}}\left( \frac{\mathbf{z}-\mathbf{Z} _{1}}{b(m)}\right) \right] =\int \frac{1}{b(m)}K _{\phi,\mathbf{x}}\left( \frac{\mathbf{z}-\mathbf{\tilde{z}}}{b(m)}\right) p _{\mathbf{z}}(\mathbf{\tilde{z}})d\mathbf{\tilde{z}}$
> >
> > where $q_{m,\phi}(\mathbf{z})$ is a kernel density estimator of $q_{\phi}(\mathbf{z}|\mathbf{x})$. You can also find the similar expression in equation (11). We simply use  expecation  with subscript 0 to denote the expecation taken w.r.t the samples from  prior distribution $p(\tilde{\mathbf{z}})$ for the sake of light notation. We hope this explanation could improve your understanding of our work.

---

> > > ### Comment · Reviewer_rQoF · 2025-03-18
> > > **Thanks for the clarifications!**
> > >
> > > My initial concerns have been addressed. Here is some additional feedback:
> > >
> > > - From my current understanding, the main motivation of the paper is to endow VAE with more flexible posterior as they are claimed to give better results. This could be made even more clear at the beginning, for instance, the new abstract writes about limitations of Gaussian latent spaces but it is not clearly mentioned in the introduction what these limitations are.  Therefore, it remains difficult for readers to understand the motivation of the work.
> > >
> > > - The paper could benefit from another pass of careful reading to check for typos and formatting, for example, two lines after Eq. (2) there is an awkward comma floating at the beginning of the line.
> > >
> > > - I am not fully convinced that closed-form of KL is essential. The first term in ELBO is approximated using MC samples, and things still work nicely. Same could be done for KL, and KL involves separable 1D integrals which can be approximated to high precision using quadrature methods. My feeling is that it should not be too difficult to implement VAE for general location scale families or mixtures. If KDE offers some advantages over these, the paper could benefit from highlighting these more clearly.

---

> > > > ### Author Response · Authors · 2025-03-18
> > > > **Thanks for the feedback!**
> > > >
> > > > Dear reviewer,
> > > >
> > > > We are glad that your initial concerns have been addressed. For additional feedback, we have updated the pdf accordingly and uploaded the newest pdf version. We list our responses as follows:
> > > >
> > > > **Response to additional feedback 1:**
> > > >
> > > > Actually, we mentioned limitations of isotropic Gaussian latent spaces through our references. For example,  we stated that "[1] and [2] attempted to replace Gaussian prior by von Mises-Fisher(vMF) distribution. The intuition is that the Gaussian distribution may have limited coverage if the true latent space is hyperspherical", "For data with heavy-tailed characteristics, [3] replaced Gaussian prior with heavy-tailed distributions such as the Student’s t-distribution, which allows the model to capture data with higher kurtosis, leading to more robust representations."
> > > >
> > > > Those works all aim to address the lack of expressibility of isotropic Gaussian latent spaces.
> > > >
> > > > [1] Abul Hasnat, Julien Bohné, Jonathan Milgram, Stéphane Gentric, and Liming Chen. von mises-fisher
> > > > mixture model-based deep learning: Application to face verification. ArXiv, abs/1706.04264, 2017
> > > >
> > > > [2] Tim R. Davidson, Luca Falorsi, Nicola De Cao, Thomas Kipf, and Jakub M. Tomczak. Hyperspherical
> > > > variational auto-encoders. ArXiv, abs/1804.00891, 2018.
> > > >
> > > > [3] Juno Kim, Jaehyuk Kwon, Mincheol Cho, Hyunjong Lee, and Joong-Ho Won. $t^3$-variational autoencoder:
> > > > Learning heavy-tailed data with student’s t and power divergence. In The Twelfth International Conference
> > > > on Learning Representations, 2024
> > > >
> > > > To help readers better understand the motivation of the work, in the updated pdf, we added the following sentence at the end of the first paragraph in introduction:
> > > >
> > > > **The isotropic Gaussian prior and posterior
> > > > distribution in vanilla VAE is mathematically convenient since the corresponding ELBO is analytic. But the
> > > > main drawbacks are the lack of expressibility of latent space and the possible posterior collapse. There are
> > > > two popular directions for extending VAEs to address these drawbacks.**
> > > >
> > > > Additionally, in the fourth paragraph of the introduction we emphasized the motivation of KDE as follows:
> > > >
> > > > "Most variants of VAE along these two directions have the closed-form KL divergence. This is essential in
> > > > implementation but somehow limits the potential applications of more flexible pairs of posterior and prior
> > > > which has no closed-form of KL divergence **but could better capture the latent data distribution and alleviate
> > > > model collapse**"
> > > >
> > > > In short, our work tries to introduce a flexible perspective in designing the latent space distribution as one edge of KDE is it's free of the distribution assumption.
> > > >
> > > > **Response to additional feedback 2:**
> > > >
> > > > We have fixed that issue and checked the typos and formatting again. We hope the updated version is of better quality.
> > > >
> > > > **Response to additional feedback 3:**
> > > >
> > > > Your intuition is correct. Actually, for complicated posterior, it's common to employ MC to approximate KL. But MC can be very time consuming for complex distributions.  However, in EVAE, the sampling step is time efficient as described by Algorithm 2.  And section 5.7 demonstrated that EVAE has a decent time efficiency compared to VAE, which is one of the advantages of EVAE.  We emphasized this point in the updated introduction (See bullet 4 in the list of main contributions)
> > > >
> > > > In addition,  implementing VAE for general location scale families or mixtures may have two obstacles: (1) For example, the number of classes in mixture Gaussian is another hyper-parameter needing extra information. (2) For quadrature methods, we assume you meant Gaussian quadrature rule.  Using quadrature methods requires the information of boundaries, i.e. lower limit and upper limit in the 1D integral. This method implicitly assumes that posterior and prior have to share the same support interval, which is restrictive in practice. What's more, if posterior and prior have different support intervals, the KL term in target function can be tricky as well.
> > > >
> > > > However, our KDE based posterior doesn't require the prior and posterior have the same support interval as the information of prior supprt is summarized by parameter B and the information of posterior is summarized by the quadratic functional I(K).  As to the mixture case, we can simply extend currenct KDE posterior to mixture KDE posterior or take higher order Taylor expansion of KL-divergence, which could be our future work.
> > > >
> > > > We hope our explanations are helpful.

---

### Decision · Action_Editor_6fco · 2025-04-10

**Recommendation:** Reject

**Comment:**

This paper proposes a method for using kernel density estimators (KDEs) as variational posteriors in VAEs. This is achieved by deriving a lower bound to the ELBO which can be analytically evaluated for the choices of prior and posterior proposed in the paper. The authors claim this is preferable over standard VAE training since it allows to use more flexible distributions than Gaussians.

Reviewers disputed the motivation behind the paper: although being able to have added flexibility over standard Gaussian VAEs is beneficial, it is unclear if simply having a better stochastic estimate of a standard ELBO would not be preferable over using an even weaker lower bound than the ELBO. Reviewers also criticized the lack of comparisons against baselines which use more flexible variational posteriors such as normalizing flows, which would be necessarily to justify the proposed approach as a desirable way of increasing the flexibility of the variational posterior. Lastly, one reviewer commented tat the writing is poor, with odd phrasing, typos, and formatting issues being present throughout the paper.

I agree with the points above and thus recommend rejection.

**Audience:**

Yes.

**Claims And Evidence:**

No, see comments below.

**Resubmission Of Major Revision:**

The authors may consider submitting a major revision at a later time.